# Testing Determinantal Point Processes

**Khashayar Gatmiry**
MIT CSAIL
gatmiry@mit.edu

**Maryam Aliakbarpour**[*]
Univ. of Massachusetts Amherst
maryam.aliakbarpour@gmail.com

**Stefanie Jegelka**
MIT CSAIL
stefje@mit.edu

## Abstract

Determinantal point processes (DPPs) are popular probabilistic models of diversity. In this paper, we investigate DPPs from a new perspective: property testing of distributions. Given sample access to an unknown distribution $q$ over the subsets of a ground set, we aim to distinguish whether $q$ is a DPP distribution, or $\epsilon$-far from all DPP distributions in $\ell_1$-distance. In this work, we propose the first algorithm for testing DPPs. Furthermore, we establish a matching lower bound on the sample complexity of DPP testing, up to logarithmic factors. This lower bound also implies a new hardness result for the problem of testing the more general class of log-submodular distributions.

## 1 Introduction

Determinantal point processes (DPPs) are a rich class of discrete probability distributions that were first studied in the context of quantum physics [54] and random matrix theory [30]. Initiated by the seminal work of Kulesza and Taskar [46], DPPs have gained a lot of attention in machine learning, due to their ability to naturally capture notions of diversity and repulsion. Moreover, they are easy to define via a similarity (kernel) matrix, and, as opposed to many other probabilistic models, offer tractable exact algorithms for marginalization, conditioning and sampling [5, 42, 46, 51]. Therefore, DPPs have been explored in a wide range of applications, including video summarization [39, 38], image search [45, 2], document and timeline summarization [53], recommendation [69], feature selection in bioinformatics [9], modeling neurons [63], and matrix approximation [22, 23, 50].

A *Determinantal Point Process* is a distribution over the subsets of a ground set $[n] = \{1, 2, \ldots n\}$, and parameterized by a *marginal kernel* matrix $K \in \mathbb{R}^{n \times n}$ with eigenvalues in $[0, 1]$, whose $(i, j)$th entry expresses the similarity of items $i$ and $j$. Specifically, the marginal probability that a set $A \subseteq [n]$ is observed in a random $\mathcal{J} \sim \mathbf{Pr}_K[.]$ is $\mathbb{P}(A \subseteq \mathcal{J}) = \det(K_A)$, where $K_A$ is the principal submatrix of $K$ indexed by $A$. This implies $\mathbb{P}(\{i, j\} \subseteq \mathcal{J}) = \det(K_{\{i,j\}}) = K_{i,i}K_{j,j} - K_{i,j}^2$ for items $i$ and $j$, which means similar items are less likely to co-occur in $\mathcal{J}$.

Despite the wide theoretical and applied literature on DPPs, one question has not yet been addressed: *Given a sample of subsets, can we test whether it was generated by a DPP?* This question arises, for example, when trying to decide whether a DPP may be a suitable mathematical model for a dataset at hand. To answer this question, we study DPPs from the perspective of *property testing*. Property testing aims to decide whether a given distribution has a property of interest, by observing as few samples as possible. In the past two decades, property testing has received a lot of attention, and questions such as testing uniformity, independence, identity to a known or an unknown given distribution, and monotonicity have been studied in this framework [18, 60].

More precisely, we ask *How many samples from an unknown distribution are required to distinguish, with high probability, whether it is a DPP or $\epsilon$-far from the class of DPPs in $\ell_1$-distance?* Given the rich mathematical structure of DPPs, one may hope for a tester that is exceptionally efficient. Yet,

---

[*]The author is also a visitor at the Simons Institute for the Theory of Computing.

we show that testing is still not easy, and establish a lower bound of $\Omega(\sqrt{N}/\epsilon^2)$ for the sample size of any valid tester, where $N = 2^n$ is the size of the domain. In fact, this lower bound applies to the broader class of *log-submodular* measures, and may hence be of wider interest given the popularity of submodular set functions in machine learning. Even more generally, the lower bound holds for testing *any* subset of log-submodular distributions that include the uniform measure.

We note that the $\sqrt{N}$ dependence on the domain size is not uncommon in distribution testing, since it is required even for testing simple structures such as uniform distributions [57]. However, achieving the optimal sample complexity is nontrivial. We provide the first algorithm for testing DPPs; it uses $\tilde{O}(\sqrt{N}/\epsilon^2)$ samples. This algorithm achieves the lower bound and hence settles the complexity of testing DPPs. Moreover, we show how prior knowledge on bounds of the spectrum of $K$ or its entries $K_{ij}$ can improve logarithmic factors in the sample complexity. Our approach relies on *testing via learning*. As a byproduct, our algorithm is the first to provably learn a DPP in $\ell_1$-distance, while previous learning approaches only considered parameter recovery in $K$ [67, 17], which does not imply recovery in $\ell_1$-distance.

In short, we make the following contributions:

- We show a lower bound of $\Omega(\sqrt{N}/\epsilon^2)$ for the sample complexity of testing any subset of the class of *log-submodular* measures which includes the uniform measure, implying the same lower bound for testing DPP distributions and strongly Rayleigh [15] measures.

- We provide the first tester for the family of DPP distributions using $\tilde{O}(\sqrt{N}/\epsilon^2)$ samples. The sample complexity is optimal with respect to $\epsilon$ and the domain size $N$, up to logarithmic factors, and does not depend on other parameters. Additional assumptions on $K$ can improve the algorithm's complexity.

- As a byproduct of our algorithm, we give the first algorithm to learn DPP distributions in $\ell_1$ distance.

## 2   Related work

**Distribution testing.** *Hypothesis testing* is a classical tool in statistics for inference about the data and model [56, 49]. About two decades ago, the framework of *distribution testing* was introduced, to view such statistical problems from a computational perspective [37, 13]. This framework is a branch of *property testing* [61], and focuses mostly on discrete distributions. Property testing analyzes the non-asymptotic performance of algorithms, i.e., for finite sample sizes. By now, distribution testing has been studied extensively for properties such as uniformity [57], identity to a known [10, 1, 25] or unknown distribution [20, 24], independence [10], monotonicity [11, 3], k-modality [21], entropy estimation [12, 70], and support size estimation [59, 68, 71]. The surveys [18, 60] provide further details.

**Testing submodularity and real stability.** Property testing also includes testing properties of functions. As opposed to distribution testing, where observed samples are given, testing functions allows an active query model: given query access to a function $f : \mathcal{X} \to \mathcal{Y}$, the algorithm picks points $x \in \mathcal{X}$ and obtains values $f(x)$. The goal is to determine, with as few queries as possible, whether $f$ has a given property or is $\epsilon$-far from it. Closest to our work in this different model is the question of testing submodularity, in Hamming distance and $\ell_p$-distance [19, 62, 31, 14], since any DPP-distribution is log-submodular. In particular, Blais and Bommireddi [14] show that testing submodularity with respect to any $\ell_p$ norm is feasible with a constant number of queries, independent of the function's domain size. The vast difference between this result and our lower bound for log-submodular distributions lies in the query model – given samples versus active queries – and demonstrates the large impact of the query model. DPPs also belong to the family of *strongly Rayleigh* measures [15], whose generating functions are real stable polynomials. Raghavendra et al. [58] develop an algorithm for testing real stability of bivariate polynomials, which, if nonnegative, correspond to distributions over two items.

**Learning DPPs.** The problem of learning DPPs has been of great interest in machine learning. Unlike testing, in learning one commonly assumes that the underlying distribution is indeed a DPP, and aims to estimate the marginal kernel $K$. It is well-known that maximum likelihood estimation for DPPs is a highly non-concave optimization problem, conjectured to be NP-hard [17, 47]. To circumvent this difficulty, previous work imposes additional assumptions, e.g., a parametric family for

$K$ [45, 44, 2, 46, 8, 48], or low-rank structure [33, 34, 29]. A variety of optimization and sampling techniques have been used, e.g., variational methods [26, 36, 8], MCMC [2], first order methods [46], and fixed point algorithms [55]. Brunel et al. [17] analyze the asymptotic convergence rate of the Maximum likelihood estimator. To avoid likelihood maximization, Urschel et al. [67] propose an algorithm based on the method of moments, with statistical guarantees. Its complexity is determined by the *cycle sparsity* property of the DPP. We further discuss the implications of their result in our context in Section 4. Using similar techniques, Brunel [16] considers learning the class of signed DPPs, i.e., DPPs that allow skew-symmetry, $K_{i,j} = \pm K_{j,i}$.

## 3 Notation and definitions

Throughout the paper, we consider discrete probability distributions over subsets of a *ground set* $[n] = \{1, 2, \ldots, n\}$, i.e., over the power set $2^{[n]}$ of size $N := 2^n$. We refer to such distributions via their probability mass function $p : 2^{[n]} \to \mathbb{R}^{\geq 0}$ satisfying $\sum_{S \subseteq [n]} p(S) = 1$. For two distributions $p$ and $q$, we use $\ell_1(q, p) = \frac{1}{2} \sum_{S \subseteq [n]} |q(S) - p(S)|$ to indicate their $\ell_1$ (total variation) distance, and $\chi^2(q, p) = \sum_{S \subseteq [n]} \frac{(q(S) - p(S))^2}{p(S)}$ to indicate their $\chi^2$-distance. Unlike the $\ell_1$-distance, the $\chi^2$-distance is a pseudo-distance, and can be lower bounded as $\chi^2(q, p) \geq 4\ell_1(q, p)^2$ by a simple application of the Cauchy-Schwarz inequality.

**Determinantal Point Processes (DPPs).** A DPP is a discrete probability distribution parameterized by a positive semidefinite kernel matrix $K \in \mathbb{R}^{n \times n}$, with eigenvalues in $[0, 1]$. More precisely, the marginal probability for any set $S \subseteq [n]$ to occur in a sampled set $\mathcal{J}$ is given by the principal submatrix indexed by rows and columns in $S$: $\mathbf{Pr}_{\mathcal{J} \sim K}[S \subseteq \mathcal{J}] = \det(K_S)$. We refer to the probability mass function of the DPP by $\mathbf{Pr}_K[J] = \mathbf{Pr}_{\mathcal{J} \sim K}[\mathcal{J} = J]$. A simple application of the inclusion-exclusion principle reveals an expression in terms of the complement $\bar{J}$ of $J$:

$$\mathbf{Pr}_K[J] = |\det(K - I_{\bar{J}})|. \tag{1}$$

**Distribution testing.** We mathematically define a *property* $\mathcal{P}$ to be a set of distributions. A distribution $q$ has the property $\mathcal{P}$ if $q \in \mathcal{P}$. We say two distributions $p$ and $q$ are $\epsilon$-*far* from ($\epsilon$-*close* to) each other, if and only their $\ell_1$-distance is at least (at most) $\epsilon$. Also, $q$ is $\epsilon$-far from $\mathcal{P}$ if and only if it is $\epsilon$-far from any distribution in $\mathcal{P}$. We define the $\epsilon$-*far set* of $\mathcal{P}$ to be the set of all distributions that are $\epsilon$-far from $\mathcal{P}$. We say an algorithm is an $(\epsilon, \delta)$-*tester* for property $\mathcal{P}$ if, upon receiving samples from an unknown distribution $q$, the following is true with probability at least $1 - \delta$:

- If $q$ has the property $\mathcal{P}$, then the algorithm outputs accept.
- If $q$ is $\epsilon$-far from $\mathcal{P}$, then the algorithm outputs reject.

We refer to $\epsilon$ and $\delta$ as *proximity parameter* and *confidence parameter*, respectively. Note that if we have an $(\epsilon, \delta)$-tester for a property with a confidence parameter $\delta < 0.5$, then we can achieve an $(\epsilon, \delta')$-tester for an arbitrarily small $\delta'$ by multiplying the sample size by an extra factor of $\Theta(\log(\delta/\delta'))$. This *amplification* technique [28] is a direct implication of the Chernoff bound when we run the initial tester $\Theta(\log(\delta/\delta'))$ times and take the majority output as the answer.

## 4 Main results

We begin by summarizing our main results, and explain more proof details in Sections 5 and 6.

**Upper bound.** Our first result is the first upper bound on the sample complexity of testing DPPs.

**Theorem 1** (Upper Bound). *Given samples from an unknown distribution $q$ over $2^{[n]}$, there exists a deterministic $(\epsilon, 0.01)$-tester for determining whether $q$ is a DPP or it is $\epsilon$-far from all DPP distributions. The tester uses*

$$O(C_{N,\epsilon} \sqrt{N}/\epsilon^2) \tag{2}$$

*samples with logarithmic factors $C_{N,\epsilon} = \log^2(N)(\log(N) + \log(1/\epsilon))$.*

Importantly, the sample complexity of our upper bound grows as $\tilde{O}(\sqrt{N}/\epsilon^2)$, which is optimal up to a logarithmic factor (Theorem 2). With additional assumptions on the spectrum and entries of $K$, expressed as $(\alpha, \zeta)$-*normal* DPPs, we obtain a refined analysis.

**Definition 1.** *For $\zeta \in [0, 0.5]$ and $\alpha \in [0, 1]$, a DPP with marginal kernel $K$ is $(\alpha, \zeta)$-normal if:*

    *1. the eigenvalues of $K$ are in the range $[\zeta, 1 - \zeta]$; and*

    *2. for $i, j \in [n] : K_{i,j} \neq 0 \Rightarrow |K_{i,j}| \geq \alpha$.*

The notion of $\alpha$-normal DPPs was also used in [67]. Since $K$ has eigenvalues in $[0, 1]$, its entries $K_{i,j}$ are at most one. Hence, we always assume $0 \leq \zeta \leq 0.5$ and $0 \leq \alpha \leq 1$.

**Lemma 1.** *For $(\alpha, \zeta)$-normal DPPs, with knowledge of $\alpha$ and $\zeta$, the factor in Theorem 1 becomes $C'_{N,\epsilon,\zeta,\alpha} = \log^2(N)(1 + \log(1/\zeta) + \min\{\log(1/\epsilon), \log(1/\alpha)\})$.*

Even more, if at least one of $\epsilon$ or $\alpha$ is not too small, i.e., if $\epsilon = \tilde{\Omega}(\zeta^{-2} N^{-1/4})$ or $\alpha = \tilde{\Omega}(\zeta^{-1} N^{-1/4})$ hold, then $C'_{N,\epsilon,\zeta,\alpha}$ reduces to $\log^2(N)$. With a minor change in the algorithm, the bound in Lemma 1 also holds for the problem of testing whether $q$ is an $(\alpha, \zeta)$-normal DPP, or $\epsilon$-far only from just the class of $(\alpha, \zeta)$-normal DPPs, instead of all DPPs (Appendix F).

Our approach tests DPP distributions via *learning*: At a high-level, we learn a DPP model from the data as if the data were generated from a DPP distribution. Then, we use a new batch of data and test whether the DPP we learnt seems to have generated the new batch of the data. More accurately, given samples from $q$, we pretend $q$ is a DPP with kernel $K^*$, and use a proper learning algorithm to estimate a kernel matrix $\hat{K}$.

But, Urschel et al. [67] derive a lower bound on the complexity of learning $K^*$ which, in the worst case, may lead to a sub-optimal sample complexity for testing. To reduce the sample complexity of learning, we do not work with a single accurate estimate $\hat{K}$, but construct a set $\mathcal{M}$ of candidate DPPs as potential estimates for $q$. We show that, with only $\Theta(\sqrt{N}/\epsilon^2)$ samples, we can obtain a set $\mathcal{M}$ such that, with high probability, we can determine if $q$ is a DPP by testing if $q$ is close to any DPP in $\mathcal{M}$. We prove that $\Theta(\log(|\mathcal{M}|)\sqrt{N}/\epsilon^2)$ samples suffice for this algorithm to succeed with high probability.

Small-scale experiments in Appendix J validate the algorithm empirically.

**Lower Bound.** Our second main result is an information-theoretic lower bound, which shows that the sample complexity of our tester in Theorem 1 is optimal up to logarithmic factors.

**Theorem 2** (Lower Bound). *Given $\epsilon \leq 0.0005$ and $n \geq 22$, any $(\epsilon, 0.01)$-tester needs at least $\Omega(\sqrt{N}/\epsilon^2)$ samples to distinguish if $q$ is a DPP or it is $\epsilon$-far from the class of DPPs.*

*Given $\alpha \in [0, 0.5]$, the same bound holds for distinguishing if $q$ is an $(\alpha, \zeta)$-normal DPP or it is $\epsilon$-far from the class of DPPs (or $\epsilon$-far from the class of $(\alpha, \zeta)$-normal DPPs).*

In fact, we prove a more general result (Theorem 4): testing whether $q$ is in any subclass $\Upsilon$ of the family of log-submodular distributions that includes the uniform distribution requires $\Omega(\sqrt{N}/\epsilon^2)$ samples. DPPs are such a subclass [46]. A distribution $f$ over $2^{[n]}$ is *log-submodular* if for every $S \subset S' \subseteq [n]$ and $i \notin S'$, it holds that $\log(f(S' \cup \{i\})) - \log(f(S')) \leq \log(f(S \cup \{i\})) - \log(f(S))$. Given the interest in log-submodular distributions [26, 66, 27, 40, 41], this result may be of wider interest. Moreover, our lower bound applies to another important subclass $\Upsilon$, *strongly Rayleigh measures* [15], which underlie recent progress in algorithms and mathematics [35, 32, 64, 4], and sampling in machine learning [5, 52, 51].

Our lower bound stands in stark contrast to the *constant* sample complexity of testing whether a given function is submodular [14], implying a wide complexity gap between access to given samples and access to an evaluation oracle (see Section 2). We prove our lower bounds by a reduction from a randomized instance of uniformity testing.

## 5 An Algorithm for Testing DPPs

We first construct an algorithm for testing the smaller class of $(\alpha, \zeta)$-normal DPPs, and then show how to extend this result to all DPPs via a coupling argument.

Our testing algorithm relies on learning: given samples from $q$, we estimate a kernel $\hat{K}$ from the data, and then test whether the estimated DPP has generated the observed samples. The magnitude

---
**Algorithm 1** DPP-TESTER
---
1: **procedure** DPP-TESTER($\epsilon$, $\delta$, sample access to $q$)
2:     $\mathcal{M} \leftarrow$ construct the set of DPP distributions as described in Theorem 3.
3:     **for** each $p$ in $\mathcal{M}$ **do**
4:         Use robust $\chi^2 - \ell_1$ testing to check if $\chi^2(q,p) \leq \epsilon^2/500$, or $\ell_1(q,p) \geq \epsilon$.
5:         **if** the tester outputs accept **then**
6:             **Return** accept.
7:     **Return** reject
---

of any entry $\hat{K}_{i,j}$ can be estimated from the marginals for $S = \{i,j\}$ and $i,j$, since $\mathbf{Pr}_K[S] = K_{i,i}K_{j,j} - K_{i,j}^2 = \mathbf{Pr}_K[i]\mathbf{Pr}_K[j] - K_{i,j}^2$. But, determining the signs is more challenging. Urschel et al. [67] estimate signs via higher order moments that are harder to estimate, but it is not clear whether the resulting $\hat{K}$ yields a sufficiently accurate estimate of the distribution to obtain an optimal sample complexity for testing. Hence, instead, we construct a set $\mathcal{M}$ of candidate DPPs such that, with high probability, there is a $\tilde{p} \in \mathcal{M}$ that is close to $q$ if and only if $q$ is a DPP. Our tester, Algorithm 1, tests closeness to $\mathcal{M}$ by individually testing closeness of each candidate in $\mathcal{M}$.

**Constructing $\mathcal{M}$.** The DPPs in $\mathcal{M}$ arise from variations of an estimate for $K^*$, obtained with $\Theta(\sqrt{N}/\epsilon^2)$ samples. Via the above strategy, we first estimate the magnitude $|K_{ij}^*|$ of each matrix entry. Separating the case $K_{ij}^* = 0$, one can compute confidence intervals for this estimation around $+|\widehat{K}_{ij}|$ and $-|\widehat{K}_{ij}|$. We then pick candidate entries from these confidence intervals, such that at least one is close to the true $K_{i,j}^*$. The candidate matrices $K$ are obtained by all possible combinations of candidate entries Since these are not necessarily valid marginal kernels, we project them onto the positive semidefinite matrices with eigenvalues in $[0,1]$. Then, $\mathcal{M}$ is the set of all DPPs parameterized by these projected candidate matrices $\Pi(K)$. Its cardinality is given in Theorem 3 and, as an explicit function of $N$ and $\epsilon$, in Appendix H.

If $q$ is a DPP with kernel $K^*$, then, by construction, our candidates contain a $\tilde{K}$ close to $K^*$. The projection of $\tilde{K}$ remains close to $K^*$ in Frobenius distance. We show that this closeness of the matrices implies closeness of the corresponding distributions $q$ and $\tilde{p} = \mathbf{Pr}_{\Pi(\tilde{K})}[.]$ in $\ell_1$-distance: $\ell_1(q,\tilde{p}) = O(\epsilon)$. Conversely, if $q$ is $\epsilon$-far from being a DPP, then it is, by definition, $\epsilon$-far from $\mathcal{M}$, which is a subset of all DPPs.

**Testing $\mathcal{M}$.** To test whether $q$ is close to $\mathcal{M}$, a first idea is to do robust $\ell_1$ identity testing, i.e., for every $p \in \mathcal{M}$, test whether $\ell_1(q,p) \geq \epsilon$ or $\ell_1(q,p) = O(\epsilon)$. But, $\mathcal{M}$ can contain the uniform distribution, and it is known that robust $\ell_1$ uniformity testing needs $\Omega(N/\log N)$ samples [68], as opposed to the optimal dependence $\sqrt{N}$.

Hence, instead, we use a combination of $\chi^2$ and $\ell_1$ distances for testing, and test $\chi^2(q,p) = O(\epsilon^2)$ versus $\ell_1(q,p) \geq \epsilon$. This is possible with fewer samples [1]. To apply this robust $\chi^2$-$\ell_1$ identity testing (described in Section 5.1), we must prove that, with high probability, there is a $\tilde{p}$ in $\mathcal{M}$ with $\chi^2(q,\tilde{p}) = O(\epsilon^2)$ if and only if $q$ is a DPP. Theorem 3, proved in Appendix A, asserts this result if $q$ is an $(\alpha,\zeta)$-normal DPP. This is stronger than its $\ell_1$ correspondent, since $4\ell_1^2(q,\tilde{p}) \leq \chi^2(q,\tilde{p})$.

To prove Theorem 3, we need to control the distance between the atom probabilities of $q$ and $\tilde{p}$. We analyze these atom probabilities, which are given by determinants, via a lower bound on the smallest singular values $\sigma_n$ of the family of matrices $\{K - I_{\bar{J}} : J \subseteq [n]\}$.

**Lemma 2.** *If the kernel matrix $K$ has all eigenvalues in $[\zeta, 1 - \zeta]$, then, for every $J \subseteq [n]$:*

$$\sigma_n(K - I_{\bar{J}}) \geq \zeta(1-\zeta)/\sqrt{2}.$$

Lemma 2 is proved in Appendix B. In Theorem 3, we observe $m = \lceil (\ln(1/\delta) + 1)\sqrt{N}/\epsilon^2 \rceil$ samples from $q$, and use the parameter $\varsigma := \lceil 200n^2\zeta^{-1}\min\{2\xi/\alpha, \sqrt{\xi/\epsilon}\}\rceil$, with $\xi := N^{-\frac{1}{4}}\sqrt{\log(n) + 1}$.

**Theorem 3.** *Let $q$ be an $(\alpha,\zeta)$-normal DPP distribution with marginal kernel $K^*$. Given the parameters defined above, suppose we have $m$ samples from $q$. Then, one can generate a set $\mathcal{M}$ of DPP distributions of cardinality $|\mathcal{M}| = (2\varsigma + 1)^{n^2}$, with $\varsigma$ defined as above, such that, with probability at least $1 - \delta$, there is a distribution $\tilde{p} \in \mathcal{M}$ with $\chi^2(q,\tilde{p}) \leq \epsilon^2/500$.*

## 5.1 Correctness of the Testing Algorithm for $(\alpha, \zeta)$-normal DPPs

Next, we show that with high probability, our resulting testing algorithm succeeds with high probability. This finishes the proof of Lemma 1. For simplicity, we set the confidence parameter in Algorithm 1 to $\delta = 0.01$. In this case, DPP-TESTER aims to output accept if $q$ is a $(\alpha, \zeta)$-normal DPP, and reject if $q$ is $\epsilon$-far from all DPPs, in both cases with probability at least 0.99.

To finish the proof for the adaptive sample complexity, we need to argue that our DPP-TESTER succeeds with high probability, i.e., that with high probability all of the identity tests, with each $p \in \mathcal{M}$, succeed. The algorithm uses robust $\chi^2$-$\ell_1$ identity testing [1], to test $\chi^2(q, p) \leq \epsilon^2/500$ versus $\ell_1(q, p) \geq \epsilon$. In our framework, the $\chi^2$-$\ell_1$ identity tester works as follows. It uses a Poissonization trick that simplifies the analysis. Given the average sample size $m$, the $\chi^2$-$\ell_1$ tester first samples $m' \sim \text{Poisson}(m)$, then obtains $m'$ samples from $q$. For each $p \in \mathcal{M}$, it then computes the statistic

$$Z^{(m)} = \sum_{J \subseteq [n]:\, p(J) \geq \epsilon/50N} \frac{(N(J) - mp(J))^2 - N(J)}{mp(J)}, \tag{3}$$

where $N(J)$ is the number of samples that are equal to set $J$, and compares $Z^{(m)}$ with the threshold $C = m\epsilon^2/10$.

Acharya et al. [1] show that for $m = \Theta(\sqrt{N}/\epsilon^2)$, $Z^{(m)}$ concentrates around its mean, which is strictly below $C$ if $p$ satisfies $\chi^2(q, p) \leq \epsilon^2/500$, and strictly above $C$ if $\ell_1(q, p) \geq \epsilon$. Let $\mathcal{E}_1$ be the event that all these robust tests, for every $p \in \mathcal{M}$, simultaneously answer correctly. To make sure that $\mathcal{E}_1$ happens with high probability, we use amplification (Section 3): while we use the same set of samples to test against every $p \in \mathcal{M}$, we multiply the sample size by $\Theta(\log(|\mathcal{M}|))$ to be confident that each test answers correctly with probability at least $1 - O(|\mathcal{M}|^{-1})$. A union bound then implies that $\mathcal{E}_1$ happens with arbitrarily large constant probability.

Theorem 3 states that, indeed, with $\Theta(\sqrt{N}/\epsilon^2)$ samples, if $q$ is an $(\alpha, \zeta)$-normal DPP, then $\mathcal{M}$ contains a distribution $\tilde{p}$ such that $\chi^2(q, \tilde{p}) \leq \epsilon^2/500$, with high probability. We call this event $\mathcal{E}_2$. DPP-TESTER succeeds in the case $\mathcal{E}_1 \cap \mathcal{E}_2$: If $q$ is an $(\alpha, \zeta)$-normal DPP, then at least one $\chi^2$-$\ell_1$ test accepts $\tilde{p}$ and consequently the algorithm accepts $q$ as a DPP. Conversely, if $q$ is $\epsilon$-far from all DPPs, then $\ell_1(q, p) \geq \epsilon$ for every $p \in \mathcal{M}$, so all the $\chi^2$-$\ell_1$ tests reject simultaneously and DPP-TESTER rejects $q$ as well. With a union bound on the events $\mathcal{E}_1^c$ and $\mathcal{E}_2^c$, it follows that $\mathcal{E}_1 \cap \mathcal{E}_2$ happens with arbitrarily large constant probability too, independent of whether $q$ is a DPP or not.

Adding the sample complexities for generating $\mathcal{M}$ and for the $\chi^2$-$\ell_1$ tests and observing $\log(|\mathcal{M}|) = O(1 + \log(1/\zeta) + \min\{\log(1/\epsilon), \log(1/\alpha)\})$ completes the proof of Lemma 1.

## 5.2 Extension to general DPPs

Next, we generalize our testing result from $(\alpha, \zeta)$-normal DPPs to general DPPs to prove the general sample complexity in Theorem 1. The key idea is that, if some eigenvalue of $K^*$ is very close to zero or one, we couple the process of sampling from $K^*$ with sampling from another kernel $\Pi_z(K^*)$ whose eigenvalues are bounded away from zero and one, i.e., parameterizing a $(0, z)$-normal DPP. This coupling enables us to test $(0, z)$-normal DPPs instead, by tolerating an extra failure probability, and transfer the above analysis for $(\alpha, \zeta)$-normal DPPs. We state our coupling argument in the following Lemma, proved in Appendix D.

**Lemma 3.** *For a value $z \in [0, 1]$, we denote the projection of a marginal kernel $K$ onto the convex set $\{A \in S_n^+ \mid zI \leq A \leq (1-z)I\}$ by $\Pi_z(K)$, where $S_n^+$ is the set of positive semidefinite matrices. For $z = \delta/2mn$, consider the following stochastic processes:*

1. *derive $m$ i.i.d samples $\{\mathcal{J}_K^{(t)}\}_{t=1}^m$ from $\mathbf{Pr}_K[.]$;*

2. *derive $m$ i.i.d samples $\{\mathcal{J}_{\Pi_z(K)}^{(t)}\}_{t=1}^m$ from $\mathbf{Pr}_{\Pi_z(K)}[.]$.*

*There exists a coupling between (1) and (2) such that*

$$\mathbf{Pr}_{coupling}\left[\{\mathcal{J}_K^{(t)}\}_{t=1}^m = \{\mathcal{J}_{\Pi_z(K)}^{(t)}\}_{t=1}^m\right] \geq 1 - \delta.$$

We can use this coupling argument as follows. Suppose the constant $c_1$ is such that using $c_1 C_{N,\epsilon,\alpha,\zeta}\sqrt{N}/\epsilon^2$ samples suffice for DPP-TESTER to output the correct answer for testing $(\alpha,\zeta)$-normal DPPs, with probability at least 0.995. Such a constant exists as we just proved. Now, we show that with $m^* = c_2 C_{N,\epsilon}\sqrt{N}/\epsilon^2$ samples for large enough constant $c_2$, we obtain a tester for the set of all DPPs. To this end, we use the parameter setting of our algorithm for $(0,\bar{z})$ normal DPPs, where $\bar{z} = 0.005/(2m^*n)$ is a function of $c_2$, $\epsilon$, and $N$. One can readily see that $c_2$ can be picked large enough, such that $m^* \geq c_1 C_{N,\epsilon,0,\bar{z}}\sqrt{N}/\epsilon^2$, with $c_2$ being just a function of $c_1$. This way, by the definition of $c_1$, the algorithm can test for $(0,\bar{z})$-normal DPPs with success probability 0.995. So, if $q$ is a $(0,\bar{z})$-normal DPP, or if it is $\epsilon$-far from all DPPs, then the algorithm outputs correctly with probability at least 0.995.

It remains to check what happens when $q$ is a DPP with kernel $K^*$, but not $(0,\bar{z})$-normal. Indeed, DPP-TESTER successfully decides this case too: due to our coupling, the product distributions $\mathbf{Pr}_{K^*}^{(m^*)}[.]$ and $\mathbf{Pr}_{\Pi_{\bar{z}}(K^*)}^{(m^*)}[.]$ over the space of data sets have $\ell_1$-distance at most 0.005, so we have $\mathbf{Pr}_{K^*}^{(m^*)}$ [Acceptance Region] $\geq \mathbf{Pr}_{\Pi_{\bar{z}}(K^*)}^{(m^*)}$ [Acceptance Region] $- 0.005 \geq 0.995 - 0.005 = 0.99$, where the last inequality follows from the fact that $\Pi_{\bar{z}}(K^*)$ is an $(0,\bar{z})$-normal DPP. Hence, for such $c_2$, DPP-TESTER succeeds with $c_2 C_{N,\epsilon}\sqrt{N}/\epsilon^2$ samples to test all DPPs with probability 0.99, which completes the proof of Theorem 1.

**Learning DPPs.** Our tester implicitly provides a method to learn a DPP $q$ in $\ell_1$-distance: the $\chi^2 - \ell_1$ tester can only accept candidate DPPs $p \in \mathcal{M}$ for which we either have $\chi^2(q,p) \leq \epsilon^2/500$ or $\ell_1(q,p) < \epsilon$. Since $\ell_1(q,p) \leq 1/2\sqrt{\chi^2(q,p)} < \epsilon$, any such $p$ is a DPP with distance $\ell_1(q,p) \leq \epsilon$ to the underlying distribution $q$. If $q$ is a DPP, we will find such a $p$ with high probability.

# 6   Lower bound

Next, we establish the lower bound in Theorem 2 for testing DPPs, which implies that the sample complexity of DPP-TESTER is tight up to logarithmic factors. In fact, our lower bound is more general: it applies to the problem of testing any subset $\Upsilon$ of the larger class of log-submodular distributions, whenever $\Upsilon$ includes the uniform measure:

**Theorem 4.** *Let $\Upsilon$ be any subset of log-submodular distributions that contains the uniform measure. For $\epsilon \leq 0.0005$ and $n \geq 22$, given sample access to a distribution $q$ over subsets of $[n]$, any $(\epsilon, 0.01)$-tester that checks whether $q \in \Upsilon$ or $q$ is $\epsilon$-far from all log-submodular distributions requires $\Omega(\sqrt{N}/\epsilon^2)$ samples.*

One may also wish to test if $q$ is $\epsilon$-far only from the distributions in $\Upsilon$. A tester for this question, however, would correctly return reject for any $q$ that is $\epsilon$-far from the set of all log-submodular distributions, and can hence distinguish the cases in Theorem 4 too. Hence, the lower bound extends to this question.

Theorem 2 is simply a consequence of Theorem 4; we may set $\Upsilon$ to be the set of all DPPs, or all $(\alpha,\zeta)$-normal DPPs. Both classes include the uniform distribution over $2^{[n]}$, which is an $(\alpha,\zeta)$-normal DPP with marginal kernel $I/2$, where $I$ is the $n \times n$ identity matrix. By the same argument, the lower bound applies to distinguishing $(\alpha,\zeta)$-normal DPPs from the $\epsilon$-far set of all DPPs for $\alpha \in [0, 0.5]$.

**Proof of Theorem 4.** To prove Theorem 4, we construct a hard uniformity testing problem that can be decided by our desired tester for $\Upsilon$. In particular, we construct a family $\mathcal{F}$, such that it is hard to distinguish between the uniform measure and a randomly selected distribution $h$ from $\mathcal{F}$. While the uniform measure is in $\Upsilon$, we will show that $h$ is far from the set of log-submodular distributions with high probability. Hence, a tester for $\Upsilon$ can, with high probability, correctly decide between $\mathcal{F}$ and the uniform measure.

We obtain $\mathcal{F}$ by randomly perturbing the atom probabilities of the uniform measure over $2^{[n]}$ by $\pm\epsilon'/N$, with $\epsilon' = c \cdot \epsilon$ for a sufficiently large constant $c$ (specified in the appendix). More concretely, for every vector $r \in \{\pm 1\}^N$ whose entries are indexed by the subsets $S \subseteq [n]$, we define the distribution $h_r \in \mathcal{F}$ as

$$\forall S \subseteq [n] : \quad h_r(S) \propto \bar{h}_r(S) = \frac{1 + r_S\epsilon'}{N},$$

where $\bar{h}_r$ is the corresponding unnormalized measure.

We assume that $h_r$ is selected from $\mathcal{F}$ uniformly at random, i.e., each entry $r_S$ is a Rademacher random variable independent from the others. In particular, it is known that distinguishing such a random $h_r$ from the uniform distribution requires $\Omega(\sqrt{N}/\epsilon'^2)$ samples [24, 57].

To reduce this uniformity testing problem to our testing problem for $\Upsilon$ and obtain the lower bound $\Omega(\sqrt{N}/\epsilon'^2) = \Omega(\sqrt{N}/\epsilon^2)$ for the sample complexity of our problem, it remains to prove that $h_r$ is $\epsilon$-far from the class of log-submodular distributions with high probability. Hence, Lemma 4 finishes the proof.

**Lemma 4.** *For $\epsilon \leq 0.0005$, $n \geq 22$ and $c = 1024$, a distribution $h_r$ drawn uniformly from $\mathcal{F}$ is $\epsilon$-far from all log-submodular distributions with probability at least 0.99.*

**Proof sketch for Lemma 4.** We fix an arbitrary log-submodular distribution $f$ and first show that (1) the $\ell_1$-distance $\ell_1(f, \bar{h}_r)$ between $f$ and the unnormalized measure $\bar{h}_r$ is large with high probability, independent of $f$ (we define the $\ell_1$-distance of general measures the same as for probability measures). Then, (2) we show that if $\ell_1(f, \bar{h}_r)$ is large, $\ell_1(f, h_r)$ is also large.

To address (1), we define a family $\mathcal{S}_r$ of subsets that, as we prove, satisfies:

(P1) With high probability, $\mathcal{S}_r$ has cardinality at least $N/64$.

(P2) For every $S \in \mathcal{S}_r$, there is a contribution of at least $\epsilon'/8N$ to $\ell_1(f, \bar{h}_r)$ from the term $V_S$ defined as

$$V_S := \tfrac{1}{2}|\bar{h}_r(S) - f(S)| + \tfrac{1}{2}|\bar{h}_r(S \cup \{1\}) - f(S \cup \{1\})| +$$
$$\tfrac{1}{2}|\bar{h}_r(S \cup \{2\}) - f(S \cup \{2\})| + \tfrac{1}{2}|\bar{h}_r(S \cup \{1, 2\}) - f(S \cup \{1, 2\})|.$$

Together, the above properties imply that $\ell_1(\bar{h}_r, f) \geq (N/64) \times (\epsilon'/8N) = \epsilon'/512$.

We define the important family $\mathcal{S}_r$ as

$$\mathcal{S}_r := \{S \subseteq [n] \setminus \{1, 2\} \mid r_{(S \cup \{1,2\})} = 1, \ r_{(S \cup \{2\})} = -1, \ r_{(S \cup \{1\})} = -1\}.$$

Property (P1) follows from a Chernoff bound for the random variables $\mathbb{1}\{S \in \mathcal{S}_r\}$, $\forall S \subseteq [n] \setminus \{1, 2\}$, where $\mathbb{1}\{.\}$ is the indicator function. For proving Property P2, we distinguish two cases, depending on the ratio $f((S \cup \{1, 2\})/f(S \cup \{2\})$. One of these cases relies on the definition of log-submodularity.

Finally, to show that (2) a large $\ell_1(f, \bar{h}_r)$ implies a large $\ell_1(f, h_r)$, we control the normalization constant $\sum_{S \subseteq [n]} \bar{h}_r(S)$. The full proof may be found in Appendix C.

# 7 Discussion

In this paper, we initiate the study of distribution testing for DPPs. Our lower bound of $\Omega(\sqrt{N}/\epsilon^2)$, where $N$ is the domain size $2^n$, shows that, despite the rich mathematical structure of DPPs, testing whether $q$ is a DPP or $\epsilon$-far from it has a sample complexity similar to uniformity testing. This bound extends to related distributions that have gained interest in machine learning, namely log-submodular distributions and strongly Rayleigh measures. Our algorithm DPP-TESTER demonstrates that this lower bound is tight for DPPs, via an almost matching upper bound of $\tilde{O}(\sqrt{N}/\epsilon^2)$.

One may wonder what changes when using the moment-based learning algorithm from [67] instead of the learner from Section 5, which yields optimal testing sample complexity. With [67], we obtain a single estimate $\hat{K}^{\text{new}}$ for $K^*$, apply a single robust $\chi^2$-$\ell_1$ test against $\mathbf{Pr}_{\hat{K}^{\text{new}}}[.]$, and return its output. The resulting algorithm DPP-TESTER2 shows a statistical-computational tradeoff: since it performs only one test, it gains in running time, but its sample complexity could be larger: Theorem 5, proved in Appendix G, states upper bounds that are is no longer logarithmic in $\alpha$ and $\zeta$, and larger than $O(\sqrt{N}/\epsilon^2)$.

**Theorem 5.** *To test against the class of $(\alpha, \zeta)$-normal DPPs, DPP-TESTER2 needs $O\left(n^4 \log(n)/\epsilon^2 \alpha^2 \zeta^2 + \ell(4/\alpha)^{2\ell} \log(n) + \sqrt{N}/\epsilon^2\right)$ samples, and runs in time $O(Nn^3 + n^6 + mn^2)$,*

*where $m$ is the number of input samples and $\ell$ is the cycle sparsity[2] of the graph corresponding to the non-zero entries of $K^*$.*

Assuming a constant cycle sparsity may improve the sample complexity, but our lower bound still applies even with assumptions on cycle sparsity.

While the results in this paper focus on sample complexity for general DPPs, it is an interesting avenue of future work to study whether additional structural assumptions, or a widening to strongly log-concave measures [6, 7], can lead to further statistical and computational benefits or tradeoffs.

## Broader Impact

Due to their ability to model negative dependencies and repulsion, DPPs have become a popular tool for modeling diversity in subset selection tasks. However, they are not the only models for negative dependence, and sometimes the decision for using DPPs may be solely based on their computational efficiency. If the true data distribution is far from being a DPP, the resulting approximation error may potentially induce biases. Our work poses the question of testing whether given data actually comes from a DPP. Being able to test for such a model fit can help avoid the biases from approximation error.

Our work provides an initial theoretical understanding of the DPP testing problem. Our results settle the general sample complexity, and open avenues for further work to improve complexity over the general baseline by identifying additional mathematical structure that may exist in the data.

### Acknowledgments

This work was supported by NSF Career award 1553284, NSF BIGDATA award IIS-1741341, NSF award CCF-1733808, CCF-1934846, NSF BIGDATA award IIS-1741137, and MIT-IBM WatsonAI Lab and Research Collaboration Agreement No. W1771646, Fintech@CSAIL. We thank Behrooz Tahmasebi, Ankur Moitra and Marwa El Halabi for helpful discussions and comments.

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
