[Supplementary Material]

# A  Proof of the Learning Guarantee

In this section, we prove Theorem 3. First, recall the definition of $(\alpha, \zeta)$-normal DPPs (Definition 1) below.

**Definition 1.** *For $\zeta \in [0, 0.5]$ and $\alpha \in [0, 1]$, a DPP with marginal kernel $K$ is $(\alpha, \zeta)$-normal if:*

1. *The eigenvalues of $K$ are in the range $[\zeta, 1 - \zeta]$; and*

2. *For $i, j \in [n] : K_{i,j} \neq 0 \Rightarrow |K_{i,j}| \geq \alpha$.*

We assume that $n$ is the size of the ground set with $N = 2^n$. We set $m = \lceil (\ln(1/\delta) + 1)\sqrt{N}/\epsilon^2 \rceil$ to be the number of samples, and use the parameter $\varsigma := \lceil 200n^2\zeta^{-1} \min\{2\xi/\alpha, \sqrt{\xi/\epsilon}\} \rceil$, with $\xi := N^{-\frac{1}{4}}\sqrt{\log(n) + 1}$. Below, we restate Theorem 3 for convenience.

**Theorem 3.** *Let $q$ be an $(\alpha, \zeta)$-normal DPP distribution with marginal kernel $K^*$. Given the parameters defined above, suppose we have $m$ samples from $q$. Then, one can generate a set $\mathcal{M}$ of DPP distributions with cardinality $|\mathcal{M}| = (2\varsigma + 1)^{n^2}$, such that, with probability at least $1 - \delta$, there is a distribution $\tilde{p} \in \mathcal{M}$ with $\chi^2(q, \tilde{p}) \leq \epsilon^2/500$.*

*Proof of Theorem 3.* To prove Theorem 3, first we estimate each entry of the marginal kernel $K^*$ and generate the set $\mathcal{M}$ of our candidate DPPs, which contains a DPP $\tilde{p} \in \mathcal{M}$ whose marginal kernel is close to $K^*$ in the Frobenius distance. Then, we show that that the closeness between the marginal kernels of $\tilde{p}$ and $q$ implies the desired upper bound in $\chi^2$-distance and $\ell_1$-distance of the two distributions. We start by introducing the initial estimate $\hat{K}$ which is obtained by estimating the entries of $K^*$ from our samples.

**Estimating entries of $K^*$:** Note that one can write the entries of the matrix $K^*$ in terms of the marginal probabilities of subsets of size one and two as follows:

$$\mathbf{Pr}_{\mathcal{J} \sim K^*}[i \in \mathcal{J}] = \det\left(\left[K_{i,i}^*\right]\right) = K_{i,i}^*, \tag{4}$$

$$\mathbf{Pr}_{\mathcal{J} \sim K^*}[\{i,j\} \subseteq \mathcal{J}] = \det\left(\begin{bmatrix} K_{i,i}^* & K_{i,j}^* \\ K_{j,i}^* & K_{j,j}^* \end{bmatrix}\right) = K_{i,i}^* K_{j,j}^* - K_{i,j}^{*}{}^2. \tag{5}$$

Given the sampled subsets $\{\mathcal{J}^{(t)}\}_{t=1}^m$, we can estimate the above marginal probabilities using the number of appearances of every single element and every pair of elements among $\mathcal{J}^{(1)}, \mathcal{J}^{(2)}, ..., \mathcal{J}^{(m)}$.

We use $\mathbb{1}E$ to denote the indicator variable of the event $E$. For each $i \in [n]$, we estimate $K_{i,i}^*$ by the average of the $\mathbb{1}\{\{i\} \subseteq \mathcal{J}^{(t)}\}$'s:

$$\hat{K}_{i,i} := \frac{1}{m}\sum_{t=1}^m \mathbb{1}\{\{i\} \subseteq \mathcal{J}^{(t)}\}.$$

We also denote the averages of the $\mathbb{1}\{\{i,j\} \subseteq \mathcal{J}^{(t)}\}$'s by $\hat{u}_{i,j}$.

$$\hat{u}_{i,j} := \frac{1}{m}\sum_{t=1}^m \mathbb{1}\{\{i,j\} \subseteq \mathcal{J}^{(t)}\}.$$

Using the estimates $\hat{u}_{i,j}$, $\hat{K}_{i,i}$, and $\hat{K}_{j,j}$, we can also estimate $K_{i,j}^{*}{}^2$ by the term $\hat{K}_{i,i}\hat{K}_{j,j} - \hat{u}_{i,j}$, based on Equation (5). To derive confidence intervals for our estimates, we use the Hoeffding bound and a union bound, which implies that with probability at least $1 - \delta$:

$$\forall i \in [n]: \quad \hat{K}_{i,i} \in \left[\mathbf{Pr}_{\mathcal{J} \sim K^*}[i \subseteq \mathcal{J}] - \xi\epsilon \,, \, \mathbf{Pr}_{\mathcal{J} \sim K^*}[i \subseteq \mathcal{J}] + \xi\epsilon\right], \tag{6}$$

$$\forall \{i,j\} \subseteq [n], i \neq j: \quad \hat{u}_{i,j} \in \left[\mathbf{Pr}_{\mathcal{J} \sim K^*}[\{i,j\} \subseteq \mathcal{J}] - \xi\epsilon \,, \, \mathbf{Pr}_{\mathcal{J} \sim K^*}[\{i,j\} \subseteq \mathcal{J}] + \xi\epsilon\right], \tag{7}$$

where $\xi := N^{-\frac{1}{4}}\sqrt{\log(n) + 1}$. Note that Equation (5) does not reveal any information about the sign of $K_{i,j}^*$. However, we can estimate its magnitude $|K_{i,j}^*|$. Thus, we consider the following two estimates for $K_{i,j}^*$:

$$\forall \{i,j\} \subseteq [n], i \neq j: \quad \begin{aligned} \hat{K}_{i,j}^{(+)} &:= \sqrt{\max\{\hat{K}_{i,i}\hat{K}_{j,j} - \hat{u}_{i,j} \,, \, 0\}}, \\ \hat{K}_{i,j}^{(-)} &:= -\sqrt{\max\{\hat{K}_{i,i}\hat{K}_{j,j} - \hat{u}_{i,j} \,, \, 0\}}. \end{aligned} \tag{8}$$

Now, let $\hat{K}_{i,j}$ be whichever of $\hat{K}_{i,j}^{(+)}$ or $\hat{K}_{i,j}^{(-)}$ that has the same sign as $K_{i,j}^*$. Then, according to Equations (6), (7), and (8), we achieve:

$$\left|\hat{K}_{i,j}^2 - K^*{}_{i,j}^2\right| \leq \left|K_{i,i}^* K_{j,j}^* - \hat{K}_{i,i}\hat{K}_{j,j}\right| + |\mathbf{Pr}_{\mathcal{J}\sim K^*}[\{i,j\} \subseteq \mathcal{J}] - \hat{u}_{i,j}|$$

$$\leq \max\{|(K_{i,i}^* + \xi\epsilon)(K_{j,j}^* + \xi\epsilon) - K_{i,i}^* K_{j,j}^*|, |(K_{i,i}^* - \xi\epsilon)(K_{j,j}^* - \xi\epsilon) - K_{i,i}^* K_{j,j}^*|\} + \xi\epsilon$$

$$\leq 3\xi\epsilon + (\xi\epsilon)^2 \leq 4\xi\epsilon,$$

where we used $\xi\epsilon \leq 1$ and that $\forall i,j \in [n] : |K_{i,j}^*| \leq 1$. Moreover, using the fact that $\hat{K}_{i,j}$ and $K_{i,j}^*$ have the same sign,

$$|\hat{K}_{i,j} - K_{i,j}^*|^2 \leq |\hat{K}_{i,j} - K_{i,j}^*||\hat{K}_{i,j} + K_{i,j}^*| = |\hat{K}_{i,j}^2 - K^*{}_{i,j}^2| \leq 4\xi\epsilon,$$

which gives

$$|\hat{K}_{i,j} - K_{i,j}^*| \leq 2\sqrt{\xi\epsilon}. \tag{9}$$

On the other hand, we have the lower bound $\alpha$ on the absolute value of the non-zero entries of $K^*$ from the $\alpha$-normality condition (1), so for non-zero $K_{i,j}^*$ we have:

$$|\hat{K}_{i,j} - K_{i,j}^*| \leq \frac{4\xi\epsilon}{|\hat{K}_{i,j} + K_{i,j}^*|} = \frac{4\xi\epsilon}{|\hat{K}_{i,j}| + |K_{i,j}^*|} \leq \frac{4\xi\epsilon}{\alpha}. \tag{10}$$

Combining Equation (10) and Equation (9), we obtain:

$$|\hat{K}_{i,j} - K_{i,j}^*| \leq 2\epsilon \min\left\{\frac{2\xi}{\alpha}, \sqrt{\frac{\xi}{\epsilon}}\right\}. \tag{11}$$

Note that by dropping the $\alpha$-normality condition, we still have the bound $|\hat{K}_{i,j} - K_{i,j}^*| \leq 2\sqrt{\xi\epsilon}$. Hence, the upper bound in Equation (11) holds even by setting $\alpha = 0$, which is equivalent to having no $\alpha$-normality for $K^*$.

**Generating candidate matrices and DPPs for $\mathcal{M}$:** Our goal is to eventually bound the $\chi^2$-distance between $q$ and our estimated distribution. To achieve this goal (as we see shortly), it is enough that one estimates each entry of $K^*$ up to an additive error of

$$\wp := \frac{\epsilon\zeta}{100n^2}. \tag{12}$$

In some natural parameter regimes, i.e. when $\epsilon = \tilde{\Omega}(\zeta^{-2}N^{-\frac{1}{4}})$ or $\alpha = \tilde{\Omega}(\zeta^{-1}N^{-\frac{1}{4}})$, $\wp$ is larger than the upper bound that we already have in Equation (11) and so we can return the distribution of $\hat{K}$ as our estimate for $q$. However, if this is not the case, we need more candidates to make sure at least one of them is close to $K_{i,j}^*$. Note that $K_{i,j}^*$ is already in the range $\left[\hat{K}_{i,j} - 2\epsilon \min\left\{2\xi/\alpha, \sqrt{\xi/\epsilon}\right\}, \hat{K}_{i,j} + 2\epsilon \min\left\{2\xi/\alpha, \sqrt{\xi/\epsilon}\right\}\right]$ with high probability. Therefore, we divide this range into $\varsigma := \lceil 2\epsilon \min\left\{2\xi/\alpha, \sqrt{\xi/\epsilon}\right\}/\wp\rceil = \lceil 200n^2\zeta^{-1}\min\{2\xi/\alpha, \sqrt{\xi/\epsilon}\}\rceil$ intervals of equal length. This way, it is guaranteed that the true $K_{i,j}^*$ is $\wp$-close to one of the midpoints of these intervals (except when $K_{i,j}^*$ is zero which we handle separately). As discussed, this partitioning (is called *bracketing* technique in the literature of learning theory) allows the algorithm to achieve the optimal sample complexity.

Now, we claim that there are $2\varsigma + 1$ candidates for $K_{i,j}^*$. This number comes from the fact that we do not know whether $\hat{K}_{i,j}$ is equal to $\hat{K}_{i,j}^{(+)}$ or $\hat{K}_{i,j}^{(-)}$ a priori. Thus, each option provides $\varsigma$ candidates. Also, we have to consider the case $K_{i,j}^* = 0$ separately because the lower bound $\alpha$ only holds for non-zero entries $K_{i,j}^*$. By considering all the combinations of candidates for each entry, we obtain a set $M$ of matrices. Since each entry has a $\wp$-close candidate, there exists a matrix $\tilde{K} \in M$ such that all of its entries are $\wp$-close to the true kernel matrix $K^*$. Therefore, this matrix is $(n\wp)$-close to $K^*$ in the Frobenius distance. As we discussed in section 5, we project each $K \in M$ onto the set of valid marginal kernels and consider the set of candidate distributions $\mathcal{M} := \{\mathbf{Pr}_{\Pi(K)}[.]| K \in M\}$. The projection, $\Pi(K)$, is with respect to the Frobenius distance between matrices, and it is easy to see that computing it is equivalent to rounding up the eigenvalues of $K$ that are negative to zero, and rounding down the ones that are greater than one to one. Now for the DPP distribution $\tilde{p} = \mathbf{Pr}_{\Pi(\tilde{K})}[.] \in \mathcal{M}$, we prove the following claims:

(C1) The kernels $\Pi(\tilde{K})$ and $K^*$ are close in operator norm:

$$\|\Pi(\tilde{K}) - K^*\|_2 \leq \frac{\epsilon\zeta}{100n}.$$

(C2) The singular values of $\Pi(\tilde{K})$ are in the range $[99\zeta/100, 1 - 99\zeta/100]$.

For the first claim (C1), it is enough to write

$$\|\Pi(\tilde{K}) - K^*\|_2 \leq \|\Pi(\tilde{K}) - K^*\|_F = \|\Pi(\tilde{K}) - \Pi(K^*)\|_F \leq \|\tilde{K} - K^*\|_F \leq n\wp = \frac{\epsilon\zeta}{100n}. \tag{13}$$

where $\|.\|_2$ and $\|.\|_F$ refer to matrix operator norm and Frobenius norm respectively. The first inequality holds because the spectral norm is bounded by the Frobenius norm, the first equality follows from the fact that $K^*$ is a valid marginal kernel, and the second inequality is because of the contraction property of projection.

Next, we prove the second claim (C2). Using the variational characterization of the Operator norm and noting the fact that $\Pi(\tilde{K}) - K^*$ is symmetric (thus its singular values are the absolute values of its eigenvalues), we have

$$\|\Pi(\tilde{K}) - K^*\|_2 = \max_{v, \|v\|_2 = 1} |v^T(\Pi(\tilde{K}) - K^*)v|.$$

Combining this with Equation (13) then implies the following for every normalized vector $\|v\|_2 = 1$:

$$-\frac{\epsilon\zeta}{100n} \leq v^T(\Pi(\tilde{K}) - K^*)v \leq \frac{\epsilon\zeta}{100n}. \tag{14}$$

Since $\mathbf{Pr}_{K^*}[.]$ is $\zeta$-normal due to our assumption, we also have

$$\zeta \leq v^T K^* v \leq 1 - \zeta. \tag{15}$$

Combining Inequalities (14) and (15) yields

$$v^T \Pi(\tilde{K}) v \geq \zeta - \frac{\epsilon\zeta}{100n} \geq \zeta - \frac{\zeta}{100} = \frac{99\zeta}{100},$$

and similarly

$$v^T \Pi(\tilde{K}) v \leq 1 - \zeta + \frac{\epsilon\zeta}{100n} \leq 1 - \frac{99\zeta}{100},$$

for any arbitrary normalized vector $v$. Finally, using the variational characterization of the smallest and largest eigenvalues, we obtain that all eigenvalues of $\Pi(\tilde{K})$ are in the range $[99\zeta/100, 1 - 99\zeta/100]$. Note that the singular values of $\Pi(\tilde{K})$ are the absolute values of its eigenvalues, simply because $\Pi(\tilde{K})$ is symmetric, which completes the proof of the second claim (C2). We use these claims (C1), (C2) in the next part.

**Closeness in parameter space implies closeness of the distributions:** In this part of the proof, we show that closeness between $K^*$ and $\Pi(\tilde{K})$ in operator norm ensures the closeness of the distributions $q$ and $\tilde{p}$ with respect to the $\chi^2$-distance and $\ell_1$-distance. This result is based on the following Lemma, whose proof we defer to the end of this section.

**Lemma 5.** *For arbitrary symmetric matrices $B$ and $E$, we have*

$$\left| |\det(B + E)| - |\det(B)| \right| \leq |\det(B)| \frac{n\|E\|_2}{\sigma_n(B)} \left( \frac{\|E\|_2}{\sigma_n(B)} + 1 \right)^{n-1},$$

*where $\sigma_n(B)$ is the smallest singular value of $B$.*

Now consider an arbitrary set $J \subseteq [n]$ and its complement $\bar{J}$. Recall that Equation (1) gives:

$$\tilde{p}(J) = |\det(\Pi(\tilde{K}) - I_{\bar{J}})|, \; q(J) = |\det(K^* - I_{\bar{J}})|.$$

Therefore, setting $B := \Pi(\tilde{K}) - I_{\bar{J}}$ and $E := K^* - \Pi(\tilde{K})$ in Lemma 5, we can upper bound $|q(J) - \tilde{p}(J)|$ as

$$|q(J) - \tilde{p}(J)| \le \tilde{p}(J) \frac{n\|E\|_2}{\sigma_n(B)} \left( \frac{\|E\|_2}{\sigma_n(B)} + 1 \right)^{n-1}. \tag{16}$$

Furthermore, from the second claim (C2) of the previous part, the singular values of $\Pi(\tilde{K})$ are in the range $[99\zeta/100, 1 - 99\zeta/100]$, which means the kernel matrix $\Pi(\tilde{K})$ satisfies the condition of Lemma 2. Therefore, from Lemma 2, the smallest singular value of $B$ is lower bounded as

$$\sigma_n(B) \ge \frac{99\zeta/100(1 - 99\zeta/100)}{\sqrt{2}} \ge \frac{99\zeta}{200\sqrt{2}},$$

where we used $1 - 99\zeta/100 > 1/2$. Combining this with the first claim (C1) of the previous part implies

$$\frac{\|E\|_2}{\sigma_n(B)} \le \frac{2\sqrt{2}\epsilon}{99n}.$$

Hence, Equation (16) gives:

$$|q(J) - \tilde{p}(J)| \le \tilde{p}(J) \frac{2\sqrt{2}\epsilon}{99} \left( \frac{2\sqrt{2}\epsilon}{99n} + 1 \right)^{n-1} \le \frac{\epsilon}{25} \tilde{p}(J), \tag{17}$$

where the last inequality follows from

$$\left( \frac{2\sqrt{2}\epsilon}{99n} + 1 \right)^{n-1} < \left( \frac{2\sqrt{2}}{99n} + 1 \right)^{n-1} < \frac{99}{50\sqrt{2}} \quad \forall n \in \mathbb{N}.$$

Note that $J \subseteq [n]$ is arbitrary, so Equation (17) finally yields the desired bound on the $\ell_1$-distance and $\chi^2$-distance between $q$ and $\tilde{p}$:

$$\ell_1(q, \tilde{p}) = \frac{1}{2} \sum_{J \subseteq [n]} |q(J) - \tilde{p}(J)| \le \sum_{J \subseteq [n]} \frac{\epsilon}{50} \tilde{p}(J) = \frac{\epsilon}{50},$$

$$\chi^2(q, \tilde{p}) = \sum_{J \subseteq [n]} \frac{(q(J) - \tilde{p}(J))^2}{\tilde{p}(J)} < \sum_{J \subseteq [n]} \frac{\epsilon^2}{500} \tilde{p}(J) = \frac{\epsilon^2}{500}.$$

$\square$

*Proof of Lemma 5.* Let $\sigma_1 \ge \cdots \ge \sigma_n$ be the singular values of $B$. For every $0 \le k \le n$, we denote $s_k$ the $k$th elementary symmetric function on the singular values of $B$, i.e.

$$s_0 = 1, \forall 1 \le k \le n : s_k = \sum_{1 \le i_1 < \ldots < i_k \le n} \sigma_{i_1} \ldots \sigma_{i_k},$$

Note that since $B$ is symmetric, the singular values are the absolute values of the eigenvalues, which implies the relation $|\det(B)| = \sigma_1 \cdots \sigma_n$.

Now Corollary 2.7 of [43] states the following determinant's perturbation inequality:

$$\left| \det(B + E) - \det(B) \right| \le \sum_{i=1}^{n} s_{n-i} \|E\|_2^i.$$

From this, we can derive

$$\left| |\det(B + E)| - |\det(B)| \right| \le \left| \det(B + E) - \det(B) \right| \le \sum_{i=1}^{n} s_{n-i} \|E\|_2^i$$

$$= |\det(B)| \sum_{i=1}^{n} \frac{s_{n-i}}{\sigma_1 \ldots \sigma_n} \|E\|_2^i,$$

where in the last equality, we multiplied and divided the sum by $|\det(B)|$. Moving forward, we bound $s_{n-i}$ by $\binom{n}{i}\sigma_1\cdots\sigma_{n-i}$:

$$
\begin{aligned}
\Big|\,|\det(B+E)| - |\det(B)|\,\Big| &\leq |\det(B)|\sum_{i=1}^{n}\binom{n}{i}\frac{\sigma_1\ldots\sigma_{n-i}}{\sigma_1\ldots\sigma_n}\|E\|_2^i \\
&= |\det(B)|\sum_{i=1}^{n}\binom{n}{i}\frac{1}{\sigma_{n-i+1}\ldots\sigma_n}\|E\|_2^i \\
&\leq |\det(B)|\sum_{i=1}^{n}\binom{n}{i}\left(\frac{\|E\|_2}{\sigma_n}\right)^i \\
&\leq |\det(B)|\,n\sum_{i=1}^{n}\binom{n-1}{i-1}\left(\frac{\|E\|_2}{\sigma_n}\right)^i \\
&= |\det(B)|\frac{n\|E\|_2}{\sigma_n}\sum_{i=0}^{n-1}\binom{n-1}{i}\left(\frac{\|E\|_2}{\sigma_n}\right)^i \\
&= |\det(B)|\frac{n\|E\|_2}{\sigma_n}\left(\frac{\|E\|_2}{\sigma_n}+1\right)^{n-1}.
\end{aligned}
$$

$\square$

# B  Uniform Lower Bound on the Smallest Singular Value of $K - I_{\bar{J}}$

In this section, we prove Lemma 2: given a marginal kernel $K$ whose eigenvalues are in the range $[\zeta, 1-\zeta]$, we prove the uniform lower bound $\zeta(1-\zeta)/\sqrt{2}$ on the singular values of the family of matrices $\{K - I_{\bar{J}}\}_{J\subseteq[n]}$. This Lemma is used in the proof of Theorem 3 and enables us to control the distances between the atom probabilities of $\mathbf{Pr}_K[.]$ and $\mathbf{Pr}_{\Pi(\tilde{K})}[.]$.

*Proof of Lemma 2.* Let $\lambda_1 \geq ... \geq \lambda_n$ be the eigenvalues of $K$ and $v_1, ..., v_n$ be an orthonormal set of their corresponding eigenvectors. We fix a subset $J \subseteq [n]$ and lower bound the smallest singular value of $K - I_{\bar{J}}$ based on its variational characterization:

$$
\sigma_n(K - I_{\bar{J}}) = \min_{\|v\|_2=1}\sqrt{v^T(K-I_{\bar{J}})^2 v}. \tag{18}
$$

Given a normalized vector $v$: $\|v\|_2 = 1$, we represent $v$ in the basis $\{v_i\}_{i=1}^n$ as $v = \sum_{i=1}^n \alpha_i v_i$. Because $\{v_i\}_{i=1}^n$ is orthonormal, we have

$$
1 = \|v\|^2 = \sum_{i=1}^{n}\alpha_i^2\|v_i\|^2 = \sum_{i=1}^{n}\alpha_i^2.
$$

Now we can express $v^T(K - I_{\bar{J}})^2 v$ as:

$$
\begin{aligned}
v^T(K-I_{\bar{J}})^2 v &= \left(\sum_{i=1}^{n}\alpha_i v_i\right)^T (K-I_{\bar{J}})^2 \left(\sum_{i=1}^{n}\alpha_i v_i\right) \\
&= \sum_{1\leq i,j\leq n}\alpha_i\alpha_j v_i^T(K-I_{\bar{J}})^2 v_j \\
&= \sum_{1\leq i,j\leq n}\alpha_i\alpha_j v_i^T K^2 v_j + \sum_{1\leq i,j\leq n}\alpha_i\alpha_j\left(v_i^T I_{\bar{J}}^2 v_j - v_i^T K I_{\bar{J}} v_j - v_i^T I_{\bar{J}} K v_j\right).
\end{aligned}
$$

Observe that $v_i^T K^2 v_i = \lambda_i^2\|v_i\|^2 = \lambda_i^2$ and $v_i^T K^2 v_j = \lambda_i\lambda_j v_i^T v_j = 0$ for $i \neq j$. We define some additional notation here: For any subset $J \subseteq [n]$, let $(v_i)_J$ be the restriction of $v_i$ into support $J$. We also denote the inner product of the vectors $v_i$ and $v_j$ restricted to $J$ by $\langle v_i, v_j\rangle_J$. Using

these notations, we can further simplify the terms $v_i^T I_{\bar{J}}^2 v_j$, $v_i^T K I_{\bar{J}} v_j$ and $v_i^T I_{\bar{J}} K v_j$ to $\langle v_i, v_j \rangle_{\bar{J}}$, $\lambda_i \langle v_i, v_j \rangle_{\bar{J}}$, and $\lambda_j \langle v_i, v_j \rangle_{\bar{J}}$ respectively. Substituting them above results in

$$
v^T (K - I_{\bar{J}})^2 v = \sum_{i=1}^n \alpha_i^2 \lambda_i^2 + \sum_{1 \le i,j \le n} (1 - \lambda_i - \lambda_j) \alpha_i \alpha_j \langle v_i, v_j \rangle_{\bar{J}}
$$

$$
= \sum_{i=1}^n \alpha_i^2 \lambda_i^2 - \sum_{1 \le i,j \le n} \alpha_i \alpha_j \lambda_i \lambda_j \langle v_i, v_j \rangle_{\bar{J}} + \sum_{1 \le i,j \le n} \alpha_i \alpha_j (1 - \lambda_i)(1 - \lambda_j) \langle v_i, v_j \rangle_{\bar{J}}
$$

where the last equality simply follows from the Equation $(1 - \lambda_i)(1 - \lambda_j) = 1 - \lambda_i - \lambda_j + \lambda_i \lambda_j$. Now substituting $\langle v_i, v_j \rangle_{\bar{J}}$ by $\langle v_i, v_j \rangle - \langle v_i, v_j \rangle_J$ in the second term above, we obtain

$$
v^T (K - I_{\bar{J}})^2 v
$$

$$
= \sum_{i=1}^n \alpha_i^2 \lambda_i^2 - \sum_{1 \le i,j \le n} \alpha_i \alpha_j \lambda_i \lambda_j \langle v_i, v_j \rangle
$$

$$
+ \sum_{1 \le i,j \le n} \alpha_i \alpha_j \lambda_i \lambda_j \langle v_i, v_j \rangle_J + \sum_{1 \le i,j \le n} \alpha_i \alpha_j (1 - \lambda_i)(1 - \lambda_j) \langle v_i, v_j \rangle_{\bar{J}}
$$

$$
= \sum_{i=1}^n \alpha_i^2 \lambda_i^2 - \sum_{i=1}^n \alpha_i^2 \lambda_i^2 + \sum_{1 \le i,j \le n} \alpha_i \alpha_j \lambda_i \lambda_j \langle v_i, v_j \rangle_J + \sum_{1 \le i,j \le n} \alpha_i \alpha_j (1 - \lambda_i)(1 - \lambda_j) \langle v_i, v_j \rangle_{\bar{J}}
$$

$$
= \left\| \sum_{i=1}^n \alpha_i \lambda_i (v_i)_J \right\|^2 + \left\| \sum_{i=1}^n \alpha_i (1 - \lambda_i)(v_i)_{\bar{J}} \right\|^2. \tag{19}
$$

Hence, it suffices to derive a lower bound on $\left\| \sum_{i=1}^n \alpha_i \lambda_i (v_i)_J \right\|^2 + \left\| \sum_{i=1}^n \alpha_i (1 - \lambda_i)(v_i)_{\bar{J}} \right\|^2$ independent from $J$. To this end, we define the column vectors $w_1 = \left( \alpha_i \lambda_i \right)_{i=1}^n$, $w_2 = \left( \alpha_i (1 - \lambda_i) \right)_{i=1}^n$. Furthermore, define $R := \left( v_1 \middle| \dots \middle| v_n \right)$ as the matrix with $v_i$ as its $i$th column, and let ${v_1'}^T, ..., {v_n'}^T$ be the rows of $R$. Because $\{v_i\}_{i=1}^n$ is an orthonormal set, $R$ is a unitary matrix, so $\{v_j'\}_{j=1}^n$ is also an orthonormal set. Next, let $V$ and $V^T$ be the subspaces spanned by the set of vectors $\{v_j'\}_{j \in \bar{J}}$ and $\{v_j'\}_{j \in J}$ respectively. Because $\{v_j'\}_{j=1}^n$ is an orthonormal set, the subspaces $V$ and $V^\perp$ are orthogonal to each other. Let $\nu_1 = \sum_{j \in \bar{J}} ({v_j'}^T w_1) v_j'$ and $\nu_1^\perp = \sum_{j \in J} ({v_j'}^T w_1) v_j'$ be the projections of $w_1$ onto $V$ and $V^\perp$ respectively. Similarly, define $\nu_2 = \sum_{j \in \bar{J}} ({v_j'}^T w_2) v_j'$ and $\nu_2^\perp = \sum_{j \in J} ({v_j'}^T w_2) v_j'$ as the projections of $w_2$ onto $V$ and $V^\perp$. Now by decomposing $w_1$ on $V$ and $V^\perp$, we can write

$$
w_1 = \nu_1 + \nu_1^\perp.
$$

Similarly, we have

$$
w_2 = \nu_2 + \nu_2^\perp.
$$

Moreover, from the orthonormality of $v_1', ..., v_n'$, we obtain

$$
\|\nu_1^\perp\|^2 = \left\| \sum_{j \in J} ({v_j'}^T w_1) v_j' \right\|^2 = \sum_{j \in J} ({v_j'}^T w_1)^2 = \sum_{j \in J} \left( \sum_{i=1}^n R_{j,i} (w_1)_i \right)^2
$$

$$
= \sum_{j \in J} \left( \sum_{i=1}^n (v_i)_j (w_1)_i \right)^2 = \left\| \sum_{i=1}^n \alpha_i \lambda_i (v_i)_J \right\|^2.
$$

Similarly, one obtains

$$
\|\nu_2\|^2 = \left\| \sum_{j \notin J} ({v_j'}^T w_2) v_j' \right\|^2 = \left\| \sum_{i=1}^n \alpha_i (1 - \lambda_i)(v_i)_{\bar{J}} \right\|^2.
$$

Combining the last two equations with Equation (19), we obtain

$$v^T(K - I_{\bar{j}})^2 v = \|\nu_2\|^2 + \|\nu_1^{\perp}\|^2. \tag{20}$$

Now, it suffices to bound $\|\nu_2\|^2 + \|\nu_1^{\perp}\|^2$. Note that

$$\|w_1\|^2 = \sum_{i=1}^{n} \alpha_i^2 \lambda_i^2 \leq \sum \alpha_i^2 = 1,$$

$$\|w_2\|^2 = \sum_{i=1}^{n} \alpha_i^2 (1 - \lambda_i)^2 \leq \sum \alpha_i^2 = 1.$$

which implies $\|\nu_1\|, \|\nu_2\|, \|\nu_1^{\perp}\|, \|\nu_2^{\perp}\| \leq 1$. Moreover, the condition $\zeta \leq \lambda_i \leq 1 - \zeta$ implies $\lambda_i(1 - \lambda_i) \geq \zeta(1 - \zeta)$. Therefore, on one hand, we get

$$\langle w_1, w_2 \rangle = \sum_{i=1}^{n} \lambda_i(1 - \lambda_i)\alpha_i^2 \geq \zeta(1 - \zeta) \sum_{i=1}^{n} \alpha_i^2 = \zeta(1 - \zeta). \tag{21}$$

On the other hand,

$$\begin{aligned}
\langle w_1, w_2 \rangle = \langle \nu_1 + \nu_1^{\perp}, \nu_2 + \nu_2^{\perp} \rangle &= \langle \nu_1, \nu_2 \rangle + \langle \nu_1^{\perp}, \nu_2^{\perp} \rangle \\
&\leq \|\nu_1\|\|\nu_2\| + \|\nu_1^{\perp}\|\|\nu_2^{\perp}\| \leq \|\nu_2\| + \|\nu_1^{\perp}\| \\
&\leq \sqrt{2(\|\nu_2\|^2 + \|\nu_1^{\perp}\|^2)} = \sqrt{2v^T(K - I_{\bar{j}})^2 v}.
\end{aligned} \tag{22}$$

where the last equality follows from Equation (20). Combining Equations (21) and (22), we conclude $v^T(K - I_{\bar{j}})^2 v \geq \zeta^2(1 - \zeta)^2/2$. Recall that $v$ is an arbitrary normalized vector, and $J$ is an arbitrary subset of $[n]$, so the variational characterization of $\sigma_n$ in Equation (18) yields the desired lower bound $\sigma_n(K - I_{\bar{j}}) \geq \zeta(1 - \zeta)/\sqrt{2}$ for every $J \subseteq [n]$. $\qquad\square$

## C  Lower Bound for Testing Log-submodular Distributions

In this section, we rigorously prove Lemma 4, which in turn completes the proof of Theorem 4. We assume that $\epsilon'$, $\mathcal{F}$, $h_r$ and $\bar{h}_r$ are defined as in Section 6.

*Detailed Proof of Lemma 4.* Given $\epsilon' \leq \frac{2}{3}$ and a log-submodular distribution $f$, we first show that the $\ell_1$-distance between $f$ and the unnormalized measure $\bar{h}_r$ is large with high probability independent of $f$ (we define the $\ell_1$-distance of general measures the same as for probability measures.) To this end, we define the following family of subsets based on $h_r$, that is random:

$$\mathcal{S}_r := \{S \subseteq [n] \setminus \{1, 2\} \mid r_{(S \cup \{1,2\})} = 1, \; r_{(S \cup \{2\})} = -1, \; r_{(S \cup \{1\})} = -1\}. \tag{23}$$

We prove that $\mathcal{S}_r$ has the following properties:

(P1) With high probability, the cardinality of $\mathcal{S}_r$ is at least $N/64$.

(P2) For every $S \in \mathcal{S}_r$, there is a contribution of at least $\epsilon'/8N$ to the $\ell_1$-distance between $\bar{h}_r$ and $f$ from the term $V_S$ defined as

$$\begin{aligned}
V_S := &\frac{1}{2}|\bar{h}_r(S) - f(S)| + \frac{1}{2}|\bar{h}_r(S \cup \{1\}) - f(S \cup \{1\})| + \\
&\frac{1}{2}|\bar{h}_r(S \cup \{2\}) - f(S \cup \{2\})| + \frac{1}{2}|\bar{h}_r(S \cup \{1, 2\}) - f(S \cup \{1, 2\})|.
\end{aligned}$$

Note that based on these two properties, one can simply derive

$$\ell_1(\bar{h}_r, f) \geq \frac{N}{64} \times \frac{\epsilon'}{8N} = \frac{\epsilon'}{512} \tag{24}$$

with high probability.

To show that the event $\mathcal{Q}_1 := \{|\mathcal{S}_r| \geq N/64\}$ happens with high probability for the first property (P1), we use a Chernoff bound for the random variables $\mathbb{1}\{S \in \mathcal{S}_r\}$, $\forall S \subseteq [n] \setminus \{1,2\}$, where $\mathbb{1}\{.\}$ is the indicator function. Clearly, for each $S \subseteq [n] \setminus \{1,2\}$, we have $\mathbb{E}[\mathbb{1}\{S \in \mathcal{S}_r\}] = \mathbf{Pr}[S \in \mathcal{S}_r] = 1/8$, and $\mathbb{E}[|\mathcal{S}_r|] = N/32$. Therefore,

$$\mathbf{Pr}[\mathcal{Q}_1^c] = \mathbf{Pr}\left[\sum_{S \in [n]\setminus\{1,2\}} \mathbb{1}\{S \in \mathcal{S}_r\} < \left(1 - \frac{1}{2}\right)\mathbb{E}[|\mathcal{S}_r|]\right] \leq \exp\left(-0.5\frac{N}{32}(\frac{1}{2})^2\right) = \exp\left(-\frac{N}{256}\right).$$

We conclude for $n \geq n_1 = 11$, $\mathcal{Q}_1$ happens with probability at least 0.995.

We now prove the second property (P2). Fix a set $S \in \mathcal{S}_r$ and define the constant $\rho := \frac{1+\epsilon'}{1-3\epsilon'/4}$. To prove $V_S \geq \frac{\epsilon'}{8N}$, we consider two cases:

**Case 1:** $\frac{f(S\cup\{1,2\})}{f(S\cup\{2\})} \leq \rho$
Here, we formalize a helper inequality in the following Lemma, and prove it at the end of this section.

**Lemma 6.** *For $a, b \geq 0$, the condition $\frac{a}{b} \leq \rho$ implies $|1 + \epsilon' - a| + |1 - \epsilon' - b| \geq \frac{\epsilon'}{4}$.*

Now from $S \in \mathcal{S}_r$, we get $\bar{h}_r(S \cup \{1,2\}) = \frac{1+\epsilon'}{N}$ and $\bar{h}_r(S \cup \{2\}) = \frac{1-\epsilon'}{N}$. Hence,

$$V_S \geq \frac{1}{2}|\bar{h}_r(S \cup \{1,2\}) - f(S \cup \{1,2\})| + \frac{1}{2}|\bar{h}_r(S \cup \{2\}) - f(S \cup \{2\})|$$

$$= \frac{1}{2}\left|\frac{1+\epsilon'}{N} - f(S \cup \{1,2\})\right| + \frac{1}{2}\left|\frac{1-\epsilon'}{N} - f(S \cup \{2\})\right| \geq \frac{\epsilon'}{8N},$$

where the last inequality follows from Lemma 6, by setting $a = Nf(S \cup \{1,2\})$, $b = Nf(S \cup \{2\})$.

**Case 2:** $\frac{f(S\cup\{1,2\})}{f(S\cup\{2\})} > \rho$
In this case, the log-submodularity property allows us to write

$$\log(f(S \cup \{1\})) - \log(f(S)) \geq \log(f(S \cup \{1,2\})) - \log(f(S \cup \{2\})) > \log(\rho),$$

or equivalently

$$\frac{f(S \cup \{1\})}{f(S)} > \rho = \frac{1+\epsilon'}{1-3\epsilon'/4}. \tag{25}$$

Note that from $S \in \mathcal{S}_r$, we have $\bar{h}_r(S \cup \{1\}) = \frac{1-\epsilon'}{N}$. If $f(S \cup \{1\})$ is larger than $\frac{1-3\epsilon'/4}{N}$, then

$$V_S \geq \frac{1}{2}|\bar{h}_r(S \cup \{1\}) - f(S \cup \{1\})| > \frac{1}{2}\left(\frac{1-3\epsilon'/4}{N} - \frac{1-\epsilon'}{N}\right) = \frac{\epsilon'}{8N}$$

and we are done. Otherwise, we have $f(S \cup \{1\}) \leq \frac{1-3\epsilon'/4}{N}$. Combining this with Equation (25) gives:

$$f(S) \leq \rho^{-1}f(S \cup \{1\}) \leq \frac{1-3\epsilon'/4}{1+\epsilon'} \times \frac{1-3\epsilon'/4}{N} \leq \frac{1-\epsilon'}{N} - \frac{\epsilon'}{4N},$$

where the last inequality follows from the condition $\epsilon' \leq \frac{2}{3}$. Finally, we obtain

$$V_S \geq \frac{1}{2}|\bar{h}_r(S) - f(S)| \geq \frac{1}{2}\left(\frac{1-\epsilon'}{N} - \left(\frac{1-\epsilon'}{N} - \frac{\epsilon'}{4N}\right)\right) = \frac{\epsilon'}{8N},$$

which completes the proof for the second property (P2). Therefore, under the occurrence of $\mathcal{Q}_1$, we conclude from Equation (24) that $\ell_1(\bar{h}_r, f) \geq \frac{\epsilon'}{512}$. To show the $\ell_1$-distance between $h_r$ and $f$ is also large, we control the normalization constant $L_r := \sum_{S\subseteq[n]} \bar{h}_r(S)$. Define the event $\mathcal{Q}_2 := \{1 - \frac{4\epsilon'}{\sqrt{N}} \leq L_r \leq 1 + \frac{4\epsilon'}{\sqrt{N}}\}$. A simple Hoeffding bound for the random variables $\frac{1+r_S\epsilon'}{N}$, $\forall S \subseteq [n]$, implies that $\mathcal{Q}_2$ happens with probability at least 0.995. Now under the occurrence

of $\mathcal{Q}_1 \cap \mathcal{Q}_2$ and assuming $n \geq n_2 = 22$, we can write:

$$2\ell_1(h_r, f) = \sum_{S \subseteq [n]} |h_r(S) - f(S)| = \sum_{S \subseteq [n]} \left| \frac{\bar{h}_r(S)}{L_r} - f(S) \right|$$

$$\geq \sum_{S \subseteq [n]} |\bar{h}_r(S) - f(S)| - \sum_{S \subseteq [n]} \bar{h}_r(S) \left| \frac{1 - L_r}{L_r} \right|$$

$$\geq \frac{\epsilon'}{256} - \frac{4\epsilon'}{L_r \sqrt{N}} \sum_{S \subseteq [n]} \bar{h}_r(S) \geq \epsilon' \left( \frac{1}{256} - \frac{4}{\sqrt{N}} \right) \geq \epsilon' \left( \frac{1}{256} - \frac{1}{512} \right) = \frac{c\epsilon}{512}.$$

A union bound for the events $Q_1^c$ and $Q_2^c$ implies that $\mathcal{Q}_1 \cap \mathcal{Q}_2$ happens with probability at least 0.99. Note that $\mathcal{Q}_1$ and $\mathcal{Q}_2$ does not depend on $f$. Setting $c = 1024$, we conclude that with probability at least 0.99, $\ell_1(h_r, f) \geq \epsilon$ for any log-submodular distribution $f$, given that $\epsilon = \epsilon'/c \leq \frac{2}{3 \times 1024}$ and $n \geq \max\{n_1, n_2\} = 22$, which completes the proof of Lemma 4. $\qquad\square$

*Proof of Lemma 6.* Here, we prove Lemma 6, which we used above. First note that if $b \geq a$, then clearly $|b - (1 - \epsilon')| + |a - (1 + \epsilon')| \geq 2\epsilon' > \frac{\epsilon'}{4}$. So we assume $b < a$.

Now define $t := a - (1 + \epsilon')$, so that $a = 1 + \epsilon' + t$. Then, we can write

$$|b - (1 - \epsilon')| + |a - (1 + \epsilon')| = |\frac{b}{a}(1 + \epsilon' + t) - (1 - \epsilon')| + |t|$$

$$\geq |\frac{b}{a}(1 + \epsilon') - (1 - \epsilon')| - |\frac{b}{a}t| + |t|$$

$$= |\frac{b}{a}(1 + \epsilon') - (1 - \epsilon')| + (1 - \frac{b}{a})|t|.$$

The condition $\frac{a}{b} \leq \rho$ implies $\frac{b}{a}(1 + \epsilon') \geq 1 - \frac{3\epsilon'}{4}$. Therefore

$$|b - (1 - \epsilon')| + |a - (1 + \epsilon')| \geq \frac{\epsilon'}{4} + (1 - \frac{b}{a})|t| \geq \frac{\epsilon'}{4}.$$

where the last inequality follows from the fact that $1 - \frac{b}{a} > 0$. $\qquad\square$

# D   Coupling DPPs

In this section, we fully introduce and prove the coupling argument of Lemma 3. Given a value $0 < z \leq 0.5$ and a DPP whose marginal kernel has eigenvalues that are outside the range $[z, 1 - z]$, the goal is to couple it with another DPP, which has a marginal kernel with all eigenvalues in $[z, 1 - z]$, such that the data sets generated from these two DPPs are equal with high probability.

*Proof of Lemma 3.* Let $V$ be an orthonormal set of the eigenvectors of $K$. For each $v \in V$, let $\lambda_v$ be its corresponding eigenvalue. To introduce our coupling, we need to define the class of *elementary DPPs* [46]. A DPP is called *elementary* if the eigenvalues of its marginal kernel are either zero or one. For each subset $V' \subseteq V$ of the eigenvectors of $K$, we consider the elementary DPP $\mathbf{Pr}_{K^{V'}}[.]$ with marginal kernel $K^{V'} := \sum_{v \in V'} vv^T$. It is well-known that any DPP can be viewed as a mixture of its corresponding elementary DPPs [46], i.e.

$$\mathbf{Pr}_K[.] = \sum_{V' \subseteq V} \left( \Pi_{v \in V'} \lambda_v \Pi_{v \notin V'} (1 - \lambda_v) \right) \mathbf{Pr}_{K^{V'}}[.]. \tag{26}$$

Using this mixture formulation, we can sample a set from $\mathbf{Pr}_K[.]$ as follows: For each eigenvector $v \in V$, we sub-sample $v$ with probability $\lambda_v$ to obtain the random subset $V'$ of $V$, then we sample $\mathcal{J}_K$ from the elementary DPP with marginal kernel $K^{V'}$. We call this sampling scheme "elementary sampling:"

- (1) For each $v \in V$, sample $y_v \sim$ Bernoulli($\lambda_v$), add $v \in V'$ if $y_v = 1$.

- (2) sample $\mathcal{J}_K \sim \mathbf{Pr}_{KV'}[.]$

According to the mixture formulation in Equation (26), the elementary sampling scheme samples $\mathcal{J}_K$ according to $\mathbf{Pr}_K[.]$.

One can readily see that the projected matrix $\Pi_z(K)$ has the same eigenvectors as $K$ but with corresponding eigenvalues $\{\bar{\lambda}_v\}_{v \in V}$, where

$$\bar{\lambda}_v = \begin{cases} \lambda_v & \text{if } \lambda_v \in [z, 1-z] \\ z & \text{if } \lambda_v < z \\ 1-z & \text{if } \lambda_v > 1-z \end{cases} \tag{27}$$

This fact follows from applying the 2-*Weilandt-Hoffman* inequality [65] for the projection operator $\Pi_z(.)$. We can similarly sample $\mathcal{J}_{\Pi_z(K)} \sim \mathbf{Pr}_{\Pi_z(K)}[.]$ with the above elementary sampling scheme. Next, we define a coupling between $\mathcal{J}_K$ and $\mathcal{J}_{\Pi_z(K)}$ as follows:

- (1) For each $v \in V$, sample $x_v \sim \text{Uniform}[0, 1]$. Then add $v$ to $V_1'$ if $x_v \in [0, \lambda_v]$, and add $v$ to $V_2'$ if $x_v \in [0, \bar{\lambda}_v]$.

- (2) if $V_1' = V_2'$, then sample $\mathcal{J} \sim \mathbf{Pr}_{KV_1'}[.]$ and set $\mathcal{J}_K = \mathcal{J}_{\Pi_z(K)} = \mathcal{J}$. Otherwise, independently sample $\mathcal{J}_K \sim \mathbf{Pr}_{KV_1'}[.]$, $\mathcal{J}_{\Pi_z(K)} \sim \mathbf{Pr}_{KV_2'}[.]$.

By looking at the marginal distributions of the sets $\mathcal{J}_K$ and $\mathcal{J}_{\Pi_z(K)}$ sampled above, we observe that $\mathcal{J}_K \sim \mathbf{Pr}_K[.]$, $\mathcal{J}_{\Pi_z(K)} \sim \mathbf{Pr}_{\Pi_z(K)}[.]$, i.e. the marginals of the coupling are as one would expect. Furthermore, if the sampled sets $V_1'$ and $V_2'$ in the first step of the sampling are equal, then $\mathcal{J}_K = \mathcal{J}_{\Pi_z(K)}$. Therefore, to lower bound $\mathbf{Pr}_{\text{coupling}}[\mathcal{J}_K = \mathcal{J}_{\Pi_z(K)}]$, it is enough to upper bound $\mathbf{Pr}_{\text{coupling}}[\mathcal{W}]$ for the event $\mathcal{W} := \{V_1' \neq V_2'\}$. But we can expand $\mathcal{W}$ as

$$\mathcal{W} = \bigcup_{v \in V} \Big( \{v \in V_1', v \notin V_2'\} \cup \{v \in V_2', v \notin V_1'\} \Big).$$

Note that for each $v \in V$, $\{v \in V_1', v \notin V_2'\} \cup \{v \in V_2', v \notin V_1'\}$ happens with probability $|\lambda_v - \bar{\lambda}_v|$. From Equation (27), we observe that $|\lambda_v - \bar{\lambda}_v| \leq z$ for every $v \in V$. Therefore, using a union bound, we obtain

$$\mathbf{Pr}_{\text{coupling}}[\mathcal{W}] \leq nz.$$

Using the definition $z = \delta/2mn$, we conclude that

$$\mathbf{Pr}_{\text{coupling}}[\mathcal{J}_K = \mathcal{J}_{\Pi_z(K)}] \geq 1 - \mathbf{Pr}_{\text{coupling}}[\mathcal{W}] \geq 1 - nz = 1 - \frac{\delta}{2m}. \tag{28}$$

Using this coupling to generate the samples $\{\mathcal{J}_K^{(t)}\}_{t=1}^m$ and $\{\mathcal{J}_{\Pi_z(K)}^{(t)}\}_{t=1}^m$, we can write

$$\mathbf{Pr}_{\text{coupling}}\Big[\{\mathcal{J}_K^{(t)}\}_{t=1}^m = \{\mathcal{J}_{\Pi_z(K)}^{(t)}\}_{t=1}^m\Big] = \Big(\mathbf{Pr}_{\text{coupling}}[\mathcal{J}_K = \mathcal{J}_{\Pi_z(K)}]\Big)^m$$

$$\geq \Big(1 - \frac{\delta}{2m}\Big)^m$$

For a real number $u$, we have the inequality

$$(1 - \frac{1}{u})^u \leq e^{-1},$$

and for $u \geq 2$, we have

$$(1 - \frac{1}{u})^u \geq e^{-\frac{u}{u-1}} \geq e^{-2}.$$

Applying these inequalities, we finally obtain

$$\mathbf{Pr}_{\text{coupling}}\Big[\{\mathcal{J}_K^{(t)}\}_{t=1}^m = \{\mathcal{J}_{\Pi_z(K)}^{(t)}\}_{t=1}^m\Big] \geq \Big(\Big(1 - \frac{\delta}{2m}\Big)^{\frac{2m}{\delta}}\Big)^{\frac{\delta}{2}} \geq e^{-\delta} \geq 1 - \delta.$$

$\square$

# E   A More Detailed Proof of Theorem 1

In this section, we take a more elaborate look at the proof of Theorem 1. The proof is mentioned in Section 5.2.

*Detailed proof of Theorem 1.* Lemma 1 tells us there exists a constant $c_1$ such that $c_1 C_{N,\epsilon,\alpha,\varsigma}\sqrt{N}/\epsilon^2$ samples suffice for DPP-TESTER to successfully test against $(\alpha,\varsigma)$-normal DPPs, with probability at least 0.995. For the general problem of testing against any DPP (i.e. without having the normality conditions), we prove that $m^* = c_2 C_{N,\epsilon}\sqrt{N}/\epsilon^2$ samples suffice to succeed with probability at least 0.99, as long as $c_2 \geq c_1 \max\{23, 2\log(c_1) + 23\}$. To test against all DPPs, we use the parameter setting of DPP-TESTER for $(0,\bar{z})$-normal DPPs, where we define $\bar{z} := 0.005/2m^*n$. The key idea is that via the coupling argument of Lemma 3, we can reduce the analysis for testing against all DPPs to the analysis for testing against only $(0,\bar{z})$-normal DPPs. To this end, we use the following Lemma. The derivation of the inequality in Lemma 7 is based on elementary algebraic operations, and we differ its proof to the end of this section.

**Lemma 7.** *For constant $c_2$ picked as large as $c_2 \geq c_1 \max\{23, 2\log(c_1) + 23\}$, we have*

$$m^* \geq C_{N,\epsilon,0,\bar{z}}\sqrt{N}/\epsilon^2. \tag{29}$$

Therefore, we pick $c_2 \geq c_1 \max\{23, 2\log(c_1) + 23\}$ to satisfy the inequality $m^* \geq C_{N,\epsilon,0,\bar{z}}\sqrt{N}/\epsilon^2$. This means that given $m^*$ samples, according to the definition of $c_1$, our tester can test against $(0,\bar{z})$-normal DPPs with success probability at least 0.995. Therefore, if the underlying distribution $q$ is an $(0,\bar{z})$-normal DPP, or if it is $\epsilon$-far from all DPPs, then DPP-TESTER outputs correctly with probability at least 0.995. It remains to show that the algorithm can also handle a DPP with kernel $K^*$, which is not $(0,\bar{z})$-normal. To see this, note that because of the particular choice of $\bar{z}$, our coupling argument in Lemma 3 implies that the product distributions $\mathbf{Pr}_{K^*}^{(m^*)}[.]$ and $\mathbf{Pr}_{\Pi_{\bar{z}}(K^*)}^{(m^*)}[.]$ over the space of data sets have $\ell_1$-distance at most 0.005. This follows from the fact that for two arbitrary random variables $X$ and $Y$ over the same underlying space, with probability distributions $P_X$ and $P_Y$, we have the following characterization of their $\ell_1$-distance:

$$\ell_1(P_X, P_Y) = \inf_{\text{coupling}(X,Y)} \mathbf{Pr}_{\text{coupling}}[X \neq Y].$$

Therefore, we have $\ell_1\left(\mathbf{Pr}_{K^*}^{(m^*)}[.], \mathbf{Pr}_{\Pi_{\bar{z}}(K^*)}^{(m^*)}[.]\right) \leq 0.005$. From this, we can relate the probability of the tester's acceptance region under $\mathbf{Pr}_{K^*}^{(m^*)}[.]$, to the same probability under $\mathbf{Pr}_{\Pi_{\bar{z}}(K^*)}^{(m^*)}[.]$:

$$\mathbf{Pr}_{K^*}^{(m^*)}[\text{Acceptance Region}] \geq \mathbf{Pr}_{\Pi_{\bar{z}}(K^*)}^{(m^*)}[\text{Acceptance Region}] - 0.005 \geq 0.995 - 0.005 = 0.99,$$

where the last inequality follows from the fact that $\mathbf{Pr}_{\Pi_{\bar{z}}(K^*)}[.]$ is an $(0,\bar{z})$-normal DPP, according to the definition of $\Pi_{\bar{z}}(K^*)$. Hence, for $c_2 \geq \max\{23, 2\log(c_1)+23\}$, DPP-TESTER, with the particular choice of its parameter $\varsigma$ with respect to $(0,\bar{z})$-normal DPPs, succeeds given $c_2 C_{N,\epsilon}\sqrt{N}/\epsilon^2$ samples to test all DPPs with probability at least 0.99. This completes the proof of Theorem 1. □

*Proof of Lemma 7.* As usual, $\log(.)$ denotes the natural logarithm. Inequality (29) boils down to

$$c_2 C_{N,\epsilon} \geq c_1 C_{N,\epsilon,0,\bar{z}},$$

or equivalently

$$c_2 \log^2(N)(\log(N) + \log(1/\epsilon)) \geq c_1 \log^2(N)(1 + \log(1/\bar{z}) + \log(1/\epsilon))$$

$$\Leftrightarrow c_2(\log(N) + \log(1/\epsilon)) \geq c_1(1 + \log(1/0.0025) + \log(m^*) + \log(n) + \log(1/\epsilon)). \tag{30}$$

Using the inequality $\log(x) \leq x - 1$ for $x > 0$, we get:

$$\log(m^*) = \log(c_2 C_{N,\epsilon}\sqrt{N}/\epsilon^2)$$

$$= \log(c_2) + 2\log(\log(N)) + \log(\log(N) + \log(1/\epsilon)) + \frac{1}{2}\log(N) + 2\log(1/\epsilon)$$

$$\leq \log(c_2) + 2(\log(N) - 1) + \log(N) + \log(1/\epsilon) - 1 + \frac{1}{2}\log(N) + 2\log(1/\epsilon)$$

$$= \log(c_2) - 2 + \frac{7}{2}\log(N) + 3\log(1/\epsilon). \tag{31}$$

Substituting Inequality (31) in Inequality (30), it is enough to satisfy

$$\frac{c_2}{c_1} \geq \frac{\log(c_2) - 1 + \log(1/0.0025) + 7/2\log(N) + 4\log(1/\epsilon) + \log(n)}{\log(N) + \log(1/\epsilon)} := \varrho.$$

We further upper bound $\varrho$ using the inequalities $\log(n) < \frac{1}{2}\log(N) + 1$ and $\log(N) \geq 0.69$:

$$\varrho < \frac{\log(c_2) + 6 + 8\log(N) + 4\log(1/\epsilon)}{\log(N) + \log(1/\epsilon)}$$

$$= \frac{\log(c_2) + 6}{\log(N) + \log(1/\epsilon)} + \frac{8\log(N) + 4\log(1/\epsilon)}{\log(N) + \log(1/\epsilon)}$$

$$\leq 1.5\log(c_2) + 9 + \frac{8(\log(N) + \log(1/\epsilon))}{\log(N) + \log(1/\epsilon)}$$

$$= 1.5\log(c_2) + 17.$$

Therefore, it is enough to satisfy $c_2/c_1 \geq 1.5\log(c_2) + 17$. But setting $c_2/c_1 = c_3$, this means we should choose $c_3$ large enough so that $c_3 \geq 1.5\log(c_3) + 1.5\log(c_1) + 17$. One can readily check that $c_3 \geq \max\{23, 2\log(c_1) + 23\}$ satisfies this inequality. Consequently, it is enough to pick $c_2$ as large as $c_2 \geq c_1\max\{23, 2\log(c_1) + 23\}$, which completes the proof of Lemma 7. Note that $c_1\max\{23, 2\log(c_1) + 23\}$ is almost a linear function of $c_1$. $\qquad\square$

## F   Modification of `DPP-Tester` for distinguishing $(\alpha, \zeta)$-normal DPPs from the $\epsilon$-far set of just the $(\alpha, \zeta)$-normal DPPs

Here, we explain how to manipulate the tester to work when we want to distinguish if $q$ is an $(\alpha, \zeta)$-normal DPP, or $\epsilon$-far only from the class of $(\alpha, \zeta)$-normal DPPs. We suggest that the reader first read the proof of Theorem 3.

The only part we change in the algorithm is the way we generate the set of candidate DPPs $\mathcal{M}$; we build the set of candidate marginal kernels $M$ the same way as in the proof of Theorem 3. Given a candidate kernel matrix $K \in M$ and an arbitrary entry $K_{i,j}$, depending on whether $K_{i,j}$ is zero, or picked from the confidence interval around $\hat{K}_{i,j}^{(+)}$ or $\hat{K}_{i,j}^{(-)}$, we define the value $\alpha_{i,j}(K)$ to be zero, $+\alpha$, or $-\alpha$ respectively. Now when we are in the case where the underlying distribution is DPP, according to the way we generate $M$, with high probability there exists a $\tilde{K} \in M$, such that $\tilde{K}_{i,j}$ is $\wp$-close to $K_{i,j}^*$ for every $i, j \in [n]$, and furthermore, $\alpha_{i,j}(\tilde{K})$ is zero if $K_{i,j}^* = 0$, or has the same sign as $K_{i,j}^*$ if $K_{i,j}^* \neq 0$ ($\wp$ is defined in Equation (12)). Our goal is to exploit this property of $\alpha_{i,j}(\tilde{K})$'s to redefine $\mathcal{M}$, so that the candidate DPPs in $\mathcal{M}$ are $(\alpha, \zeta)$-normal. To this end, for each matrix $K \in M$, instead of projecting $K$ onto the set of PSD matrices with eigenvalues in $[0, 1]$, we project onto the following convex body with respect to the Frobenius distance, which is a subset of $(\alpha, \zeta)$-normal DPPs:

$$D_K := \{A \in S_n^+ | \; \zeta.I \preceq A \preceq (1 - \zeta)I, \, \forall i, j \in [n] :$$
$$A_{i,j}/\alpha_{i,j}(K) \geq 1 \text{ if } \alpha_{i,j}(K) \neq 0, \text{ or } A_{i,j} = 0 \text{ if } \alpha_{i,j}(K) = 0\},$$

and generate $\mathcal{M}$ as

$$\mathcal{M} := \{\mathbf{Pr}_{\Pi_{D_K}(K)}[.] | \, K \in M\},$$

where we denote by $\Pi_{D_K}$ the projection map onto $D_K$. Particularly, it is clear that $D_K$ is a subset of $(\alpha, \zeta)$-normal DPPs, and as the intersection of convex sets, $D_K$ is also convex, so projection on $D_K$ is well-defined.

Now when $q$ is a DPP with marginal kernel $K^*$, we know it is $(\alpha, \zeta)$-normal, so for every $i, j \in [n]$ : $|K_{i,j}^*| \geq \alpha$. Combining this with the property that $\alpha_{i,j}(\tilde{K})$ is zero if $K_{i,j}^* = 0$, or it has the same sign as $K_{i,j}^*$ if $K_{i,j}^* \neq 0$, we obtain that $K^* \in D_{\tilde{K}}$. This means $\Pi_{D_{\tilde{K}}}(K^*) = K^*$. Using this relation with the contraction property of projection, we obtain

$$\|\Pi_{D_{\tilde{K}}}(\tilde{K}) - K^*\|_F = \|\Pi_{D_{\tilde{K}}}(\tilde{K}) - \Pi_{D_{\tilde{K}}}(K^*)\|_F \leq \|\tilde{K} - K^*\|_F.$$

Therefore, by substituting the projection $\Pi(K)$ in our algorithm by $\Pi_{D_K}(K)$ for every $K \in M$, the inequality in Equation (13) in the proof of Theorem 3 remains to hold, and the rest of the proof for

the $\chi^2$-distance bound follows accordingly. On the other hand, with the new projection $\Pi_{D_K}(K)$ instead of $\Pi(K)$, the DPPs that are generated in $\mathcal{M}$ are all $(\alpha, \zeta)$-normal, so if we are in the case that $q$ is $\epsilon$-far from $(\alpha, \zeta)$-normal DPPs, it is also $\epsilon$-far from $\mathcal{M}$. Consequently, our $\chi^2$-$\ell_1$ tests are able to distinguish the two cases as before, and we obtain an $(\epsilon, 0.99)$-tester with sample complexity $\Theta(\sqrt{N}/\epsilon^2)$ for this modified version of our testing problem.

We should note that computing $\Pi_{D_K}(K)$ is trickier than $\Pi(K)$; for $\Pi(K)$, computing the Singular value decomposition (SVD) of $K$ is enough (or we can use iterative algorithms to get an approximate solution faster), but computing $\Pi_{D_K}(K)$ is a general convex problem and is solvable via convex programming approaches.

## G  Analysis of `DPP-Tester2`

In this section, we show the argument in Theorem 5, which is a direct consequence of the sample and time complexities for the moment-based learning algorithm in [67].

*Proof of Theorem 5.* Recall from the proof of Theorem 3 that estimating each entry of $K^*$ up to accuracy $\wp$, defined in Equation (12), is enough to prove the desired bound $\chi^2(q, \tilde{p}) \le \epsilon^2/500$, which in turn enables the final $\chi^2$-$\ell_1$ tester to work correctly.

Now let $\mathcal{D}_n$ be the set of $n \times n$ diagonal matrices with $+1$ or $-1$ on their diagonal. For any $D \in \mathcal{D}_n$, the marginal kernel $DK^*D$ induces the same DPP distribution as $K^*$ does. In other words, $K^*$ is identifiable only up to the multiplication of its rows and columns by $\pm 1$. With this in mind, to get the final guarantee for closeness of the DPP distributions when we use the moment-based learning algorithm, i.e. $\chi^2\left(q, \mathbf{Pr}_{K^{\text{new}}}[.]\right) \le \epsilon^2/500$, it is enough that for some $D \in \mathcal{D}_n$, we estimate the matrix $DK^*D$ entrywise with accuracy $\wp$. In fact, the moment-based learning algorithm gives us such a guarantee; according to [67], in order to compute a $\wp$-accurate estimate of $K^*$ in *pseudo-distance*, the moment-based algorithm requires $O\left( \left(\frac{1}{\alpha^2 \wp^2} + \ell(\frac{4}{\alpha})^{2\ell}\right) \log(n) \right)$ samples, where the pseudo-distance of matrices $K_1$ and $K_2$ is defined as

$$\rho(K_1, K_2) = \min_{D \in \mathcal{D}_n} \left| DK_1 D - K_2 \right|_\infty = \min_{D \in \mathcal{D}_n} \max_{i,j \in [n]} \left| (DK_1 D)_{i,j} - (K_2)_{i,j} \right|.$$

Now substituting $\wp$ from Equation (12), the sample complexity of the moment-based algorithm as a subroutine in `DPP-Tester2` becomes

$$m = O\left( \frac{n^4 \log(n)}{\epsilon^2 \alpha^2 \zeta^2} + \ell(\frac{4}{\alpha})^{2\ell} \log(n) \right), \tag{32}$$

where $\ell$ is the cycle sparsity[3] of the graph with vertices $[n]$, whose edges correspond to the non-zero entries of $K^*$.

Adding the complexity of the final $\chi^2$-$\ell_1$ test to the learning complexity in Equation (32), the overall sample complexity of `DPP-Tester2` is:

$$O\left( \frac{n^4 \log(n)}{\epsilon^2 \alpha^2 \zeta^2} + \ell(\frac{4}{\alpha})^{2\ell} \log(n) + \frac{\sqrt{N}}{\epsilon^2} \right).$$

For the time complexity, the run-time of the moment-based algorithm is $O(n^6 + mn^2)$ in the worst-case due to [67], and the run-time of the $\chi^2$-$\ell_1$ test is $O(Nn^3 + m)$, as we have to compute $\mathbf{Pr}_{K^{\text{new}}}[J]$ for each $J \subseteq [n]$, requiring an SVD in time $O(n^3)$. Adding them up results in an overall run time of

$$O(Nn^3 + n^6 + mn^2) = O(\epsilon^4 m^2 n^3 + n^6 + mn^2) = \text{Poly}(m, n)$$

for `DPP-Tester2`, where the above equality follows from our sample complexity lower bound $m = \Omega(\sqrt{N}/\epsilon^2)$. $\square$

## H  Time complexity of `DPP-Tester`

In this section, we analyze the time complexity of `DPP-Tester`.

For each $p \in \mathcal{M}$, to apply the robust $\chi^2$-$\ell_1$ test of Acharya et al. [1], one has to compute the statistic $Z^{(m)}$ defined in Equation (3). To compute $Z^{(m)}$, one should compute $p(J)$ for every $J \subseteq [n]$, which requires a determinant calculation in time $O(n^3)$. Therefore, each robust $\chi^2 - \ell_1$ testing takes time $O(Nn^3)$. There is another $O(m)$ pre-processing time for computing $N(J)$'s. Moreover, computing the projection matrix $\Pi(K)$ for every $K \in M$ requires the Singular value decomposition (SVD) of $K$, which takes time $O(n^3)$. This is because we project with respect to the Frobenius distance, and it follows from the 2-*Weilandt-Hoffman* inequality [65] that computing $\Pi(K)$ can equivalently be done by rounding down the eigenvalues of $\hat{K}$ that are larger than one to one, and rounding up the eigenvalues that are negative to zero. Computing the initial estimate of the marginal kernel, i.e. $\hat{K}$ in the proof of Theorem 3, also takes time at most $O(\min\{N,m\}n^2)$. Therefore, the overall time complexity becomes

$$O(|\mathcal{M}|Nn^3 + m).$$

To have a time complexity upper bound only in terms of the main variables $n, \epsilon$, note that based on what was discussed in section 5.2, for the general DPPs without the knowledge of $\zeta$ and $\alpha$, we set the normality parameters in our algorithm as $(\alpha, \zeta) = (0, \bar{z})$, where $\bar{z}$ is $0.005/(2m^*n)$, for $m^* = O(C_{N,\epsilon}\sqrt{N}/\epsilon^2)$. Substituting $C_{N,\epsilon} = \log^2(N)(\log(N) + \log(1/\epsilon))$, we get that $\bar{z}^{-1} = O\left((n^4 + n^3 \log(1/\epsilon))\sqrt{N}/\epsilon^2\right)$. Substituting $\zeta = \bar{z}$ in Theorem 3, in the definition of $\varsigma$ and ignoring $\alpha$ in the $\min$ term, we obtain the following worst-case scenario upper bound on $\varsigma$:

$$\varsigma = O((n^2\zeta^{-1}\sqrt{\xi/\epsilon}) = O\left(n^2(n^4 + n^3 \log(1/\epsilon))\sqrt{N}/\epsilon^2)N^{-\frac{1}{8}}\log^{\frac{1}{4}}(n)\epsilon^{-0.5}\right) \quad (33)$$

$$= O\left(\epsilon^{-2.5}(n^6 + n^5 \log(1/\epsilon))N^{\frac{3}{8}}\log^{\frac{1}{4}}(n)\right). \quad (34)$$

Therefore,

$$|\mathcal{M}| = O\left(\epsilon^{-2.5}(n^6 + n^5 \log(1/\epsilon))N^{\frac{3}{8}}\log^{\frac{1}{4}}(n)\right)^{n^2}.$$

But notice that our matrices are symmetric, hence, we only have to consider different candidates for at most $n(n+1)/2$ entries, which reduces the size of $|\mathcal{M}|$ to

$$|\mathcal{M}| = O\left(\epsilon^{-2.5}(n^6 + n^5 \log(1/\epsilon))N^{\frac{3}{8}}\log^{\frac{1}{4}}(n)\right)^{n(n+1)/2}.$$

## I  Lower bound on the Sample Complexity of Distinguishing the Uniform distribution from $\mathcal{F}$

In this section, we give a high-level sketch of the approach that Diakonikolas and Kane [24] use, to argue a lower bound of $\Omega(\sqrt{N}/\epsilon^2)$ on the sample complexity of the problem of testing the uniform distribution against $h_r$, randomly selected from $\mathcal{F}$.

*Proof.* Suppose that we observe samples from the underlying distribution $g$, where $g$ can either be $h_r$ or the uniform distribution. We flip a random coin $X$, and based on that set $g$ to the uniform distribution, or to $h_r$, a distribution randomly selected from $\mathcal{F}$. For every $S \subseteq [n]$, let $N(S)$ be the number of samples that are equal to $S$. We aim to show that given the number of samples satisfy $m = o(\sqrt{N}/\epsilon^2)$, the information in the collection of random variables $\mathcal{A} = \{N(S) | S \subseteq [n]\}$ is not enough to guess the value of $X$ strictly better than random guessing, say with success probability greater than $0.51$.

To begin, we use the following Lemma without proof, which is exactly Lemma 3.2. in page 19 of [24]. This is a classical result in Information theory:

**Lemma 8.** *For random variables $X$ and $\mathcal{A}$, if there exist a function mapping $\mathcal{A}$ to $X$ such that $f(\mathcal{A}) = X$ with probability at least $0.51$, then we have the following bound on their mutual information:*

$$I(X; \mathcal{A}) \geq 2.10^{-4}.$$

Based on Lemma 8, it is enough to show that $I(X; \mathcal{A}) = o(1)$. To continue, we use the Poissonization trick; instead of directly deriving $m$ samples from $g$, we sample $m'$ from the Poisson distribution with parameter $m$, namely $m' \sim \text{Poisson(m)}$, then derive $m'$ samples from $g$. Using this trick, we still have $m' = \Theta(m)$ samples with high probability, so it is enough to bound $I(X, \mathcal{A})$ for $\mathcal{A}$ with respect to the new sampling scheme with Poissonization. Based on properties of the Poisson distribution, the new scheme is equivalent to deriving $N(S) \sim \text{Poisson}(mg(S))$ for each set $S \subseteq [n]$ independent from the others. Furthermore, we showed in the proof of Theorem 4 that $L_r = \Theta(1)$ with high probability, so by using $mL_r$ instead of $m$ samples, the order of sample size does not change. But now, in the case $g = \bar{h}_r$, $N(S)$ is sampled according to $N(S) \sim \text{Poisson}(mL_r h_r(S)) = \text{Poisson}(m\bar{h}_r(S))$. Thus, one can readily see that again, we can substitute $h_r$ by its unnormalized counterpart $\bar{h}_r$ in our Poisson sampling.

Finally, assuming the sampling scheme $N(S) \sim \text{Poisson}(m\bar{h}_r(S))$, $\forall S \subseteq [n]$, we bound $I(X, \mathcal{A})$. Note that given the value of $X$, the random variables $\{N(S)\}$ are independent, so we have the following bound on the mutual information:

$$I(X; \mathcal{A}) \le \sum_{S \subseteq [n]} I(X; N(S)). \tag{35}$$

It is enough to bound each of the terms $I(X; N(S))$. For that, we bring without proof Lemma 3.3. from [24], page 20:

**Lemma 9.** *If $N(S) \sim Poisson(m\bar{h}(S))$ for $X = 0$ and $N(S) \sim Poisson(m/N)$ for $X = 1$, then:*

$$I(X; N(S)) = O(m^2 \epsilon^4 / N^2).$$

From this Lemma and Equation (35), we get $I(X; \mathcal{A}) = o(m^2 \epsilon^4 / N) = o(1)$. Combining this with Lemma 8, we conclude that we need at least $\Omega(\sqrt{N}/\epsilon^2)$ samples to non-trivially guess $X$ from the observed samples. This completes the proof of the promised lower bound on the sample complexity of the problem of testing uniform distribution against $\mathcal{F}$. For more details and the proof of Lemmas 8 and 9, we refer the reader to [24]. □

## J  Experiments

Finally, we perform small-scale synthetic experiments as a proof of concept.

We generate random DPPs for $n = 4$ by randomly generating kernel matrices $K$. We draw the eigenvalues of each $K$ uniformly from $[0, 1]$, and use eigenvectors of random matrices with entries uniformly sampled from $[0, 1]$. To generate a $\Theta(\epsilon)$-far distribution from the class of DPPs, we use our lower bound approach in section 6: we add a random perturbation of $\pm \frac{\epsilon}{N}$ to each atom probability of the uniform distribution over $2^n$. Lemma 4 implies that for sufficiently large $n$ and small $\epsilon$, with high probability, we are $\Theta(\epsilon)$ far from the class of DPPs, where the constant in $\Theta(\epsilon)$ is in the range $[1/1024, 1]$. Since we do not know the exact value of this constant, we use the constant $1/2$ to compute the algorithm's acceptance threshold: $C = m(\frac{\epsilon}{2})^2/10$.

We simplified the algorithm slightly in two ways: (1) instead of projecting the candidate matrices, we just ignore the ones that have an eigenvalue outside the range $[0, 1]$; (2) Instead of checking multiple candidate entries in the confidence intervals for each $K_{ij}^*$, we only consider the two signed values $+|\widehat{K}_{ij}|$ and $-|\widehat{K}_{ij}|$. The results are obtained by averaging the empirical probabilities over 20 runs.

Figure 1 shows the performance of our tester for various number of samples: detection rate when the underlying distribution is a DPP (blue bars), and False Alarm rate when it is $\Theta(\epsilon)$ far from the class of DPPs (orange bars). For $\epsilon = 0.02$, and the $C$ we picked here, the algorithm correctly accepts most DPPs, but needs more samples to correctly reject non-DPPs.

Our adaptive sample complexity has a weak logarithmic dependence on $\zeta^{-1}$; as a reminder, $\zeta$ measures how close the eigenvalues of $K$ are to zero or one. The coupling argument in Lemma 3 got rid of this dependence, for $\zeta$ below some threshold. This theory motivates the question how much the accuracy of our tester depends on the spectrum of $K$, in particular, on the distribution of its eigenvalues. To investigate this for $n = 4$, we sample the eigenvalues of $K$ from a normal distribution with mean on one of the equidistant points $0.05, 0.1, \ldots, 0.9, 0.95$ and standard deviations $0.1$ or $0.2$, conditioned on the interval $[0, 1]$.

Figure 1: Detection and False Alarm rates of the testing algorithm for various numbers of samples and $\epsilon = 0.02$.

Figure 2: Detection errors of the testing algorithm for DPP kernel matrices with eigenvalues sampled from a conditional normal distribution, with different means, variances, and over multiple choices of the algorithm's threshold $C$.

Figure 2 shows the results for a variety of parameters. The $x$-axis is the mean of the normal distribution, while the $y$-axis is the empirical value of the error probability in Detection (i.e. recovering the underlying DPP), averaged over 100 runs for each setting of the parameters. The sample size is 10000 here. The results suggest that the detection accuracy is only very weakly affected by the mean of the eigenvalues of $K$ and, in particular, the error does not increase a lot at the boundaries.

## Footnotes

[3]The cycle sparsity of a graph is the smallest $\ell'$ such that the cycles with length at most $\ell'$ constitute a basis for the cycle space of the graph.