[Reviews · NeurIPS 2020]

Review 1

Summary and Contributions: Fix some positive integer n and let D denote the family of all subsets of {1, …, n}. Determinantal Point Processes (DPP) induce a set of probability distributions supported on D. Given i.i.d. samples from an unknown distribution P supported on D, can we distinguish whether P comes from a DPP or it is “epsilon-far” from being a DPP? This paper gives lower and upper bounds for the number of samples needed to achieve this task, which are within logarithmic factors of one another. Namely, the paper proves the optimal sample complexity for testing in the total variation distance is on the order of 2^(n/2)/epsilon^2 times poly (n) times polylog(1/epsilon).

Strengths: There has been lots of recent interest in DPPs and testing them is a natural problem, which had not been studied before. This paper proves nearly optimal bounds for this problem. The algorithmic techniques are new and interesting.

Weaknesses: Some parts need better explanations.

Correctness: I haven’t checked the details but the ideas seem robust.

Clarity: Yes.

Relation to Prior Work: Yes.

Reproducibility: Yes

Additional Feedback: Thanks for the rebuttal. I haven't changed my score. At the end of Section 1 you say you give the first algorithm for learning DPPs. Yet this is not elaborated on in the paper, there is no theorem quantifying this sentence formally. Chi-squared distance is not super well-known, so some explanation and its relationship with the L1 distance is needed. Line 148—149: “the algorithm … has a lower bound on sample complexity” does not make sense. Note that algorithms do not have lower bounds, problems do. You meant “the algorithm … needs this many samples” perhaps. Lemma 4: Explain what “with high probability” means here. Explain why the uniform distribution corresponds to a DPP. In the abstract, it must be mentioned that the lower and upper bounds differ by logarithmic factors. Strictly speaking, they are not “matching.” Line 40—41: the sentence “reveals a large gap between sample access and query access” is unclear at this point of the paper. Line 188: the sentence “pick candidate entries from the confidence intervals around plus minus | K_{i,j} |” is not clear. You mean you consider both plus K_{i,j} and minus K_{i,j}? What confidence interval? In Section 5.2, is the main idea that the algorithm practically cannot tell whether the data is coming from K* or the “polished” version of K*? If yes, adding such an informal sentence will help the reader understand what’s happening. Comment on the running time of your algorithm. Is it polynomial in the input size? Please add in the paper that Theorem 5 is proved in Appendix G.


Review 2

Summary and Contributions: This paper presents an approach for performing property testing for determinantal point processes (DPPs). That is, given samples from an unknown distribution over subsets of a ground set, the aim is to determine whether the unknown distribution is a DPP, or epsilon-far from the class of DPPs in l_1 distance. A lower bound is obtained for the sample complexity of testing any subset of a log-submodular distribution, including a DPP. The first testing algorithm for the family of DPP distributions is presented. Finally, an algorithm for learning DPPS in both l_1 and chi^2 distances is presented.

Strengths: - To the best of my knowledge, this paper appears to be the first work on property testing for DPPs, and opens up new territory for future work on this topic. - The testing algorithms and bounds on sample complexity presented in this work appear to be well grounded from a theoretical perspective. - The most expensive term in the DPP-Tester2 algorithm is n^6, which indicates that this algorithm is only feasible for scenarios involving relatively small ground sets. This is a limitation of this algorithm. However, the runtime of this algorithm is much more feasible than that of the DPP-Tester algorithm (see weaknesses, below).

Weaknesses: - While this paper makes strong theoretical contributions, no experiments are conducted. It would be very helpful to validate the theory presented in this paper by running experiments on small, synthetically-generated datasets. - The lower bound of sqrt(N)/(epsilon^2) on the number of samples needed to test if a distribution is a DPP or epsilon-far from the class of DPPs, while theoretically sound, suggests that DPP testing is likely infeasible in practice for a moderately-sized ground set (e.g., a ground set of 100 items or more), since N is the size of the power set of the ground set. That is, an infeasibly large number of samples would need to be collected and checked. - The most expensive term in the time complexity of the DPP-Tester algorithm is \Mcal N n^3. From Alg. 1 (DPP-Tester) and Theorem 3, the cardinality of \Mcal (the size of the set of constructed DPPs) is (2 \zeta + 1)^(n^2), where n is the size of the ground set, and \zeta \in [0, 0.5]. This suggests that this algorithm can only be feasibly run for DPPs with very small ground sets (e.g., n = 10), which is a significant limitation of this algorithm.

Correctness: The claims and methodology appear to be correct. Note that no experimental results presented in this paper. Substantial details are presented in the paper, and supplement, regarding proofs, analysis, and explanations for the theoretical results in this work.

Clarity: The paper is reasonably well written.

Relation to Prior Work: A clear overview of prior work on distribution testing, property testing, and DPP learning is presented in Section 2 of the paper.

Reproducibility: Yes

Additional Feedback: For Alg. 1 (DPP-Tester), the cardinality of \Mcal (the size of the set of constructed DPPs) is (2 \zeta + 1)^(n^2), where n is the size of the ground set, and \zeta \in [0, 0.5]. It will quickly become infeasible to generate this set of DPPs as the size of the ground set grows to even small ground sets (e.g., n = 20). Do the authors have some ideas on how to reduce the cardinality of \Mcal to make this algorithm more feasible in practice? -- Post-rebuttal update: I've read the other reviews and the authors' rebuttal. I'm satisfied that the rebuttal has addressed the questions and comments from the reviewers, and hope to see an improved version of this paper at the conference, including results for synthetic experiments. I maintain my overall score of a 7.


Review 3

Summary and Contributions: EDIT: I've read the rebuttal and I am satisfied with the answers. Assume one is given access to m samples of a random set of points taken from a finite ground set of size n. The authors investigate statistical procedures that tests whether the underlying random set is a DPP. A lower bound on the number of samples m is given, by reduction of the problem to a known testing problem, and an algorithm is given that matches the lower bound. The lower bound is a rather negative result for ML applications, since m should be at least exponential in the size n of the ground set.

Strengths: Given the growing number of ML applications of DPPs and related measures showing negative dependence, the topic is impactful. The paper is well written and seems rigorous, although I wasn't able to check all proofs in detail for lack of time. While the problem could be better motivated application-wise, testing DPPs and related measures is definitely a natural problem from a statistical point of view. Moreover, I believe that the fact that testing for DPPs is as hard as testing for uniformity is surprising enough to warrant publication. The learning method used in the proof is too brute force to be useful per se, in my opinion, but this doesn't diminish the importance of the theoretical result. In other words, I believe that this paper establishes an interesting baseline for a new problem.

Weaknesses: My only (minor) concern is with the style of the paper, which should be clarified in a few places; see main comments below.

Correctness: yes

Clarity: clarity could be improved in a few places; see comments below.

Relation to Prior Work: yes

Reproducibility: Yes

Additional Feedback: # Major comments * L42 maybe restate here that sqrt(N) is exponential in the size n of the ground set, thus making testing so far impractical. At first read, I had forgotten that N was not the size of the ground set, and I was thus optimistic about the sqrt(N) dependence. Overall, denoting N=2^n is confusing, as N is also commonly used as the size of the ground set for DPPs, for instance in the seminal [Kulesza and Taskar, ref. 46]. * L115 throughout the document, there are a few occurrences of the phrasing "with probability at least ..." where it is not completely clear what probability measure is considered. Here, do you mean probability over the samples from q, or probability over a possibly randomized tester, or both? Now that I've read the paper, I am guessing that your testers are deterministic, but at this stage I wasn't sure. * L148-150 "But"..."testing." This passage is unclear: what do you mean? * L186 you should say a few words on the cardinality of the set \mathcal{M}. From Appendix A, I gather that this set comes from discretizing the Cartesian product of n^2 intervals, so that its cardinality is exponential in n. * L288 Proof of theorem 4: this passage is slightly unclear while this reduction argument seems key to the paper. For example L292, does the "high probability" refer to the uniform measure over F, to the samples from the resulting h, or both? Also, one has to wait until L300 to understand why you cook up this construction, that is, you want to use the hardness result from [24,57]. I would quote a mathematical statement of the result from [24,57] and then explain in detail how you are going to derive a tester for their problem. Although, can you explain why it is needed to randomize h? * Do you believe that your negative results extend to DPPs over arbitrary spaces, say over R^d? Even without thinking about DPPs, is there a similar impossibility of testing uniformity in the continuous case? * Out of curiosity, assume you do not have access to an exact sampler, but to an approximate sampler, say within a controlled total variation distance of the target distribution. Can you extend your results and test whether this target distribution is a DPP? # Minor comments * L24 "Hence": this may be a bit jumping to the conclusion for readers not familiar with to DPPs. Maybe just write the 2x2 determinant giving the probability of co-occurrence to explain what you mean. * L41 maybe explain that "query access" means pointwise evaluation of the pdf. * L104 Insist here that you are restricting to DPPs with symmetric kernels (it is implicit since you assume K to be positive semi-definite, but I would say it aloud), and that you are thus using a theorem by Macchi-Soshnikov that characterizes the DPPs with symmetric kernels. * Appendix A: maybe restate the overall notation before starting the appendices: n, N, m, etc. * L145 "as if the data is" -> "as if the data were" * L180 determining * L252 "delta/2mn" maybe disambiguate the notation here, do you mean delta/(2mn)?


Review 4

Summary and Contributions: This work tackles for the first time the non-trivial task of DPP testing. This is done via a learning strategy to estimate the entries of the marginal kernel K combined with confidence intervals characterizing a set M of candidate DPPs. Then a robust chi2-l1 test between the target distribution and all candidate DPPs is performed. The authors prove a 2^n / eps^2 lower bound on the sample complexity for DPP testing. An extension of this result to log-submodular distributions is provided.

Strengths: To the best of my knowledge, this is the first work focused on DPP testing, which is a non-trivial and important question to tackle. The lower bound on the sample complexity obtained for DPP testing extends the more general log-submodular distributions. The rich bibliography may be helpful for further research on DPP testing.

Weaknesses: The paper is very technical with a lot of notations making reading difficult. There is no illustration of the methodology, appropriate figures would certain help the reader to build a mental map of the method. While the ultimate purpose of testing is practical, the paper presents no experiments. In particular is not clear at all how to implement l4 of Algorithm 1. A simple toy example would surely help illustrate the methodology and support the mathematical derivations. The lower bound on the sample complexity is exponential in the size n of the ground set and the size of the class of candidate DPPs : this makes DPP testing all but practical.

Correctness: While the methodology seems correct, there are quite a few technical developments in the paper and it is impossible to verify every piece given the short turnaround. I read some proofs which are correct.

Clarity: The paper is well written but also very technical making reading difficult. This work may fit better a journal format. Some illustrations of the methodology are expected.

Relation to Prior Work: Yes it is.

Reproducibility: Yes

Additional Feedback: In this work, general symmetric DPP testing is considered, how easier would it be to focus on testing against a parametric class of DPPs? How could summary statistics of the samples, e.g., number of points, and more general linear statistics sum_{j in \calJ} f(j) be used in DPP testing strategies? Could you give some more insights on the sqrt(N)/eps^2 lower bound for testing the uniform distribution? It seems that the log-submodularity is the intrinsic property which yields the sqrt(N)/eps^2 lower bound, do you have any insights on the role played by negative dependence property? The notion of identifiability of the kernel K, or equivalence class of kernels does not seem to be used to define the candidate class M, is it a implicit consideration? Typo: l214 the constant 'x' in |M| = (2'x' + 1)^n^2 is not defined. **EDIT** I am satisfied with the authors' response, which was clear and honest. I expect the camera-ready to address the reviewers comments on the writing, but also to show some experimental simulations validating their testing by learning procedure.

[Author Response · NeurIPS 2020]

We thank all reviewers for their positive comments.

**Reviewer #1**  **Learning:** Our tester implicitly learns, too: To learn a DPP, we can run the tester and return any DPP
in the candidate set $\mathcal{M}$ that is accepted by the $\chi^2$-$\ell_1$ test (if any). By Theorem 1, with high probability, the returned
DPP is close to the DPP whose samples we observe. We will make this more explicit. **Lemma 4:** "high probability"
means with an arbitrarily large constant probability. We will clarify this. **Uniform distribution:** In Line 286 we state
why the uniform measure is a DPP. We will clarify this. **Confidence Interval:** The tester uses $|\hat{K}_{i,j}|$ as an estimate
for $|K^*_{i,j}|$ (the sign is harder to estimate). For some parameter $u$, concentration assures that w.h.p. $|\hat{K}_{i,j}|$ belongs to
$(|K^*_{i,j}| - u, |K^*_{i,j}| + u)$. The algorithm picks multiple equally-distant values from the intervals $(|\hat{K}_{i,j}| - u, |\hat{K}_{i,j}| + u)$
and $(-|\hat{K}_{i,j}| - u, -|\hat{K}_{i,j}| + u)$. We will clarify this. **Section 5.2:** The main idea is to use the coupling to bound the
acceptance probability of the tester for $\pi_{\bar{z}}(K^*)$ instead of $K^*$, by applying Lemma 1, and then use their closeness to
transfer the result. We will clarify this. **Run time:** is not generally polynomial as $|\mathcal{M}|$ is not polynomially bounded by
$n$. We will add this.
**Other comments:** We will add/clarify these and correct the typos. Thank you.

**Reviewer #2**  We will add synthetic data experiments in the supplement. As the reviewer correctly pointed out, our
exponential lower bound leaves no hope for a more efficient tester. But we hope that it motivates follow-up works
studying structural assumptions, sub- and super-classes that may allow better results. **Reducing the order of $|\mathcal{M}|$:** We
are not aware of any systematic way for this. Yet, a simple idea is to consider the identifiability classes of $K$: for each
$i \in [n]$, multiplying the $i$th row and column of the marginal kernel by $-1$ does not change the distribution. This defines
an equivalence relation between matrices, where each matrix is equivalent to $2^{n-1}$ others. Currently, we are considering
all of the $2^{n-1}$ matrices of a class in $\mathcal{M}$. Also, as a heuristic method, one can substitute the discrete search in $\mathcal{M}$ by an
iterative approach: We are looking for a DPP in $\mathcal{M}$ for which the statistic $Z^{(m)}$ is less than $C$. One can see $Z^{(m)}$ as a
smooth function of the kernel $K$, and minimize it using Gradient Descent (GD) with initialization $\hat{K}$.

**Reviewer #3**  **L42:** We will clarify this. **L115:** Indeed, our tester is deterministic. Except the statement of choosing a
random distribution in $\mathcal{F}$, all high probability statements, including the one in L115, are w.r.t. $q$, whose samples we
observe. **L148-150:** This is a typo. We meant that using their algorithm results in sub-optimal sample complexity for
testing. We will correct this. **L186:** We will add a note on the cardinality of $\mathcal{M}$. **L288:** The "high probability" in L292
refers only to the randomness of selecting $h$ from $\mathcal{F}$. We will explain this. Randomization is necessary for the hardness
result: compared to any fixed distribution in $\mathcal{F}$, randomization further decreases the $\ell_1$ distance to the uniform measure,
which makes testing harder. **Testing continuous DPPs:** testing in continuous space without any parameterization is
generally infeasible with finite samples. For example, consider $\epsilon$-uniformity testing over $[0, 1]$: One can divide $[0, 1]$
into $N$ sub-intervals and consider only distributions that assign constant density to each. This discretization transfers the
lower bound $\sqrt{N}/\epsilon^2$ of the discrete problem to the continuous case, for any $N$. We conjecture that a similar negative
result holds for testing continuous DPPs. **Approximate Sampler:** We are currently not sure if our results generalize to
an approximate sampler. The tester starts by estimating $K^*$ by computing the marginal probabilities for every $i \in [n]$
and $\{i, j\} \subseteq [n]$. An approximate sampler with $\ell_1$ error $e$ adds an additional error term of $e$ to the confidence intervals
of our estimates for these marginals, and propagates to our learning guarantee. However, we do not currently know how
the robust identity tests might behave with an approximate sampler. We thank the reviewer for mentioning this useful
generalization, as oftentimes an MCMC sampler is used for DPPs. **Minor comments:**  Thank you, we will follow
these.

**Reviewer #4** We will make the writing more accessible, add illustrations, and include synthetic experiments. **L4 in**
**the algorithm:** In section 5.1, we briefly explain what the $\chi^2 - \ell_1$ test does. We will make it clearer. **Testing against**
**parametric DPPs:** Indeed, our results motivate a study of additional structural assumptions that may bypass the lower
bound. Currently, we do not know which sub-classes may enable this. **Summary Statistics:** Indeed, one hopes to
exploit the parametric structure of DPPs to design good summary statistics for testing. We use the marginals in the
learning part (based on the DPP structure), and the $\chi^2$-$\ell_1$ tester computes $Z^{(m)}$. Our information-theoretic lower
bound also guarantees that there are no better statistics for $\ell_1$ testing. **Uniformity testing intuition:** Intuitively, the
dependence on $\epsilon$ comes from Central limit theorem: To test if a coin is unbiased or has bias $\epsilon$ with the sum statistic, at
least $\Omega(1/\epsilon^2)$ samples are needed. For the intuition of the dependence on $N$, consider this problem: we observe samples
from either the uniform distribution over $2N$, or over $N$ items. By to the Birthday Paradox, with $o(\sqrt{N})$ samples,
most likely no repetition of samples happens. But, without that, the two distributions are intuitively indistinguishable.
**Negative Dependence:** Indeed, our lower bound relies only on submodularity, and does not directly generalize to every
class with the negatively dependence property. **Identifiability of K:** Indeed, during the generation of $\mathcal{M}$, we do not
classify different kernel representations of a DPP. Reducing the repetitions reduces the size of $\mathcal{M}$.

[Meta-Review · NeurIPS 2020]

The paper was unanimously recognized as a strong contribution and I suggest a strong accept. All the reviewers also requested some improvement in writing which I expect the authors to do.